# ReDi: Rectified Discrete Flow

**Jaehoon Yoo**
KAIST
wogns98@kaist.ac.kr

**Wonjung Kim**
KAIST
wjhj16@kaist.ac.kr

**Seunghoon Hong**
KAIST
seunghoon.hong@kaist.ac.kr

## Abstract

Discrete Flow-based Models (DFMs) are powerful generative models for high-quality discrete data but typically suffer from slow sampling speeds due to their reliance on iterative decoding processes. This reliance on a multi-step process originates from the factorization approximation of DFMs, which is necessary for handling high-dimensional data. In this paper, we analyze the factorization approximation error using Conditional Total Correlation (TC), and reveal its dependence on the coupling. To address the challenge of efficient few-step generation, we propose Rectified Discrete Flow (ReDi), a novel iterative method that reduces the underlying factorization error (measured as Conditional TC) by rectifying the coupling between source and target distributions. We theoretically prove that each ReDi step guarantees a monotonic decreasing Conditional TC, ensuring its convergence. Empirically, ReDi significantly reduces Conditional TC and enables few-step generation. Moreover, we demonstrate that the rectified couplings are well-suited for training efficient one-step models on image generation. ReDi offers a simple and theoretically grounded approach for tackling the few-step challenge, providing a new perspective on efficient discrete data synthesis. Code is available at https://github.com/Ugness/ReDi_discrete.

## 1 Introduction

Discrete data synthesis is a fundamental task across many domains, including texts, images, and biological sequences. Recent advances in deep generative models have shown remarkable success in synthesizing high-quality data [2, 7, 29, 39]. Among these, Discrete Flow-based Models (DFMs) [1, 5, 6, 13, 26, 30, 31, 34, 35], which typically achieve this by modeling a probabilistic process that transforms a simple initial state (e.g., a masked or random state) into complex data, have emerged as particularly effective for this purpose, demonstrating strong performance in generating high-quality discrete data.

However, despite their quality, the reliance of DFMs on a slow, iterative multi-step sampling process [1, 5, 6, 13] prohibits efficient few-step generation. This reliance originates from factorization approximations applied to model high-dimensional discrete data [17]. While data often exhibits high inter-dimensional correlation, the factorization approximation assumes independence of dimensions given the previous state. This assumption becomes increasingly inaccurate for the large steps required in few-step generation, undermining DFMs' effectiveness in few-step scenarios.

To alleviate the slow sampling of DFMs, existing approaches [10, 17, 31] often involve techniques like knowledge distillation where a multi-step teacher model trains a few-step student model. These methods commonly require maintaining both teacher and student models simultaneously during training and may introduce new, specialized training objectives distinct from standard DFM training, adding complexity.

In this paper, we address the few-step generation challenge in DFMs by analyzing the underlying factorization error. We characterize this error using Conditional Total Correlation (TC) [38] and

reveal its dependence on the coupling. Inspired by Rectified Flows [24, 25, 42] in continuous domain, we propose Rectified Discrete Flow (ReDi) to enable efficient few-step generation by rectifying the coupling of discrete data, which in turn reduces the Conditional TC. By focusing on coupling rectification, our method provides a simpler alternative to prior works [10, 17, 31], as it requires neither a specialized training strategy nor the handling of separate teacher-student models, which in turn reduces memory requirements. This simplicity enables ReDi to be broadly applicable to various DFMs including other distillation frameworks.

We demonstrated our method's effectiveness both theoretically and empirically. We theoretically prove that each ReDi iteration guarantees monotonically decreasing Conditional TC and empirically show that each rectification significantly reduces it. We evaluated our method on class conditional image generation and text generation. On image generation, ReDi shows comparable few-step generation performance against existing distillation methods, and significantly outperforms in one-step generation, due to direct rectification of the couplings contributing to the factorization error. On text generation, we observe that iteratively applying rectification improves the sampling efficiency and that ReDi can also be applied with existing distillation methods.

## 2 Related Works

### 2.1 Discrete Flow-based Models

Discrete flow-based models (DFMs) are used to generate discrete data such as images [2, 7, 8, 16, 39], videos [43, 44], text [1, 29–31, 34, 35], and protein [6]. DFMs generate data by learning the flow from initial states (often set as masked states or uniform random states). The generative flow is learned by two primary formalisms, reversing a corruption process (e.g., masked generative models [2, 7, 8, 39], discrete diffusion [1, 26, 30, 34, 35]), or constructing bridges between initial distribution and data distribution (e.g., Discrete Flow Matching [6, 13], Schrödinger Bridges [21, 22]). Although they show powerful performance on discrete data synthesis, and efficient sampling cost compared to autoregressive models as they support generating multiple states simultaneously [10, 30, 31, 34], they still require slow multi-step decoding process for successive generation [17].

### 2.2 Distillation of Discrete Flow-based Models

To address the slow sampling speeds of multi-step DFMs, prior works [10, 17, 31] have explored methods for distillation and faster generation. These approaches typically aim to distill a slower, multi-step teacher model into a faster, few-step student model. They primarily focus on modifying the training objective or designing specific training procedures tailored for distillation, and are sometimes specific to a particular DFM framework. For instance, SDTT [10] is tailored for masked diffusion models, and the dual consistency distillation method suggested in DUO [31] is tailored for uniform diffusion. While Di4C [17] suggested an objective function that is applicable for various discrete diffusion models, it utilizes four loss terms, requiring tuning of weights.

Alternative approaches [23, 41] introduce auxiliary models to reduce decoding steps. Discrete Copula Diffusion [23], for instance, requires a pretrained autoregressive model as an additional copula model. EDLM [41] takes another approach, using an energy-based model to guide sampling; however, its practical sampling efficiency remains limited as its algorithm requires sampling multiple candidates and selecting the most probable one. In contrast to the prior works, our method improves few-step generation in DFMs by focusing on the coupling itself, rather than solely on modifying the training process or model architecture.

### 2.3 Rectified Flows on Continuous Data

Rectified Flow [24] and related techniques [20, 25, 42] represent a significant development in achieving efficient few-step generation for flow-based models in the continuous domain. These methods address the limitations of standard ODE solvers for faster sampling by characterizing the error as non-straightness of the path and proposing techniques to rectify or straighten the flow defined by the coupling between distributions. While Rectified Flow [24] is popular and well-used techniques in the continuous domain, the rectification of discrete flows has not been explored prior to this work. This is primarily because the concept of straightness is difficult to define meaningfully in a discrete space. Furthermore, the core challenge for few-step generation in DFMs stems not from

continuous-time path straightness, but from the factorization approximation inherent in modeling high-dimensional discrete data. In this paper, we bridge this gap by characterizing this factorization error using Conditional Total Correlation.

## 3 Method

This section describes our proposed method, Rectified Discrete Flow (ReDi). We first provide preliminary background on Discrete Flow-based Models (DFMs) in Sec. 3.1. Then we detail the problem of factorization error, which is characterized by Conditional Total Correlation (TC), depends on the coupling or dataset in Sec. 3.2. Finally, we present our novel method, ReDi, which rectifies the coupling to reduce factorization error in Sec. 3.3. We also provide the theoretical analysis that a single step of rectification process monotonically reduces the Conditional TC.

### 3.1 Preliminary: Discrete Flow-based Models

We denote discrete data as a sequence of discrete random variables $X = (X^1, X^2, \cdots, X^N)$, where each dimension $X^i$ takes values from a set of size $D$. The primary objective of DFMs [1, 5, 6, 13, 26, 30, 34] is to learn the probabilistic mapping from the source distribution $p(X_0)$ to the target distribution $q(X_1)$ by modeling the conditional distribution $p(X_1|X_0)$. For data generation, the source distribution $p(X_0)$ is typically chosen as a tractable distribution (*e.g.*, uniform).

To achieve this transformation, DFMs define a probability path over time $t \in [0, 1]$ as a marginal distribution $p_t(x_t) = \mathbb{E}_{(X_0, X_1) \sim \pi}[p_t(x_t|X_0, X_1)]$. This path is defined using a coupling $\pi$ and a conditional probability distribution $p_t(x_t|x_0, x_1)$ that specifies the path bridging specific states $x_0$ and $x_1$. The coupling $\pi$ is a joint distribution over $(X_0, X_1)$ from which training pairs are drawn, which is commonly chosen as an independent coupling i.e., $\pi(x_0, x_1) = p(x_0)q(x_1)$. A common choice for $p_t(x_t|x_0, x_1)$ is a convex sum bridging the two endpoints. For instance,

$$p_t(x_t|x_0, x_1) = (1 - \alpha_t)\delta_{x_0}(x_t) + \alpha_t\delta_{x_1}(x_t), \tag{1}$$

where $\alpha_t$ is a time-dependent coefficient with $\alpha_0 = 0$ and $\alpha_1 = 1$ and $\delta$ is a Kronecker delta function, *i.e.*, $\delta_{x_0}(x_t) = 1$ if $x_0 = x_t$ and 0 otherwise.

DFMs are trained to model the conditional transition probability $p_{s|t}(x_s|x_t)$ for $s > t$. These transitions are mathematically derived from the defined probability path and describe the probabilistic dynamics of moving from a state $X_t$ to $X_s$ along the path as:

$$p_{s|t}(x_s|x_t) = \sum_{x_0, x_1} p_s(x_s|x_0, x_1, x_t)p_t(x_0, x_1|x_t), \tag{2}$$

where $p_s(x_s|x_0, x_1, x_t)$ is the conditional probability of $X_s = x_s$ given the path passes through $x_t$ at time $t$ and has endpoints $x_0, x_1$. Also, $p(x_0, x_1|x_t)$ is the posterior distribution over endpoints given $X_t = x_t$ and is computed using Bayes' rule:

$$p_t(x_0, x_1|x_t) = \frac{p_t(x_t|x_0, x_1)\pi(x_0, x_1)}{\sum_{x'_0, x'_1} p_t(x_t|x'_0, x'_1)\pi(x'_0, x'_1)}. \tag{3}$$

As both terms on the right side of Eq. 2 are ultimately derived from the probability path definition and the coupling $\pi$, the conditional transition probability $p_{s|t}(x_s|x_t)$ directly depends on $\pi$. By learning to model these fundamental step-wise transitions, DFMs acquire the ability to simulate the entire path and thereby transform samples from the source distribution to the target distribution.

However, modeling the full conditional distribution $p_{s|t}(x_s|x_t)$ over the entire $D^N$ state space is intractable for high-dimensional data. Therefore, to make modeling feasible, DFMs assume that the true transition $p_{s|t}(x_s|x_t)$ in Eq. 2 can be approximated by a factorization across dimensions:

$$p_{s|t}(x_s|x_t) \approx \prod_{i=1}^{N} p_{s|t}(x_s^i|x_t), \tag{4}$$

where $p_{s|t}(x_s^i|x_t)$ is the marginal distribution of $p_{s|t}(x_s|x_t)$ along dimension $i$. By relying on this factorization, DFMs reduce the complexity of the output space representation. While this factorization is necessary to deal with high-dimensional data, the resulting approximation error may induce a few-step decoding challenge, which we discuss in detail in the following.

$\pi_0(X_0, X_1)$

| $X_0$ \ $X_1$ | 00 | 01 | 10 | 11 | $p(X_0)$ |
|---|---|---|---|---|---|
| 00 | $\frac{1}{8}$ | | | $\frac{1}{8}$ | 1/4 |
| 01 | $\frac{1}{8}$ | | | $\frac{1}{8}$ | 1/4 |
| 10 | $\frac{1}{8}$ | | | $\frac{1}{8}$ | 1/4 |
| 11 | $\frac{1}{8}$ | | | $\frac{1}{8}$ | 1/4 |
| $p(X_1)$ | 1/2 | 0 | 0 | 1/2 | |

$\pi_1(X_0, X_1)$

| $X_0$ \ $X_1$ | 00 | 01 | 10 | 11 | $p(X_0)$ |
|---|---|---|---|---|---|
| 00 | $\frac{1}{4}$ | | | | 1/4 |
| 01 | $\frac{1}{4}$ | | | | 1/4 |
| 10 | | | | $\frac{1}{4}$ | 1/4 |
| 11 | | | | $\frac{1}{4}$ | 1/4 |
| $p(X_1)$ | 1/2 | 0 | 0 | 1/2 | |

Figure 1: A synthetic example that illustrates two different couplings $\pi_0$ and $\pi_1$. $p(X_0)$ is defined as a uniform distribution over $\{00, 01, 10, 11\}$ and $p(X_1)$ is defined as a uniform distribution over $\{00, 11\}$. While the two couplings $\pi_0$ and $\pi_1$ share the same marginal distributions ($p(X_0)$ and $p(X_1)$), due to the difference between them, the Conditional Total Correlation of $\pi_0$ is higher than that of $\pi_1$. Detailed explanation about the example is in Sec. 3.2.

### 3.2 Factorization Error of DFMs

Factorization error induced by the approximation in Eq. 4 hinders few-step generation in DFMs. Specifically, since the factorized model treats dimensions independently, it fails to create the inter-dimensional correlation needed as the state changes from uncorrelated $X_0$ to correlated $X_1$. This error grows with the time step $\Delta = s - t$, as larger steps involve more significant changes in the distribution being modeled, making few-step generation difficult.

We characterize the factorization error using Conditional TC, which is defined as the expected KL divergence between the conditional distribution and the product of its marginals:

$$TC_\pi(X_s|X_t) = \mathbb{E}_{x_t}\Big[ D_{KL}\Big( p_{s|t}(X_s|X_t = x_t) || \prod_{i=1}^{N} p_{s|t}(X_s^i|X_t = x_t) \Big) \Big]. \tag{5}$$

This metric is particularly suitable for analyzing the factorization error because it directly quantifies the inter-dimensional dependencies that the factorized approximation neglects. It also shows that several prior distillation objectives [10, 17, 31] implicitly minimize Eq. 5 by reducing the KL divergence between the multi-step teacher transition (serving as the approximated joint distribution) and the few-step student transition. Importantly, we note that the above TC is dependent on coupling $\pi$, since the true transition $p_{s|t}(x_s|x_t)$ depends on $\pi$ as shown in Eq. 2 and Eq. 3. This dependency reveals that rectifying the coupling can reduce the Conditional TC, which characterizes the factorization error.

To provide further intuition, we present a simple example in Fig. 1 explaining the dependency between coupling and the factorization error. For simplicity, we consider a task that models the distribution of 2-bit sequences, where $p(X_0)$ is defined as a uniform distribution over 4 states $\{00, 01, 10, 11\}$ and $p(X_1)$ is defined as a uniform distribution over $\{00, 11\}$. We then compare the two couplings $\pi_0$ and $\pi_1$, which leads to the same marginal distributions $p(X_0)$ and $p(X_1)$ but different Conditional TC. For instance, for the coupling $\pi_0$, $p(X_1 = 00|X_0 = 00) = 0.5$ diverges with its factorized distribution $p(X_1^1 = 0|00)p(X_1^2 = 0|00) = 0.25$, while the factorization error is zero for the coupling $\pi_1$ by $p(X_1 = 00|00) = p(X_1^1 = 0|00)p(X_1^2 = 0|00) = 1$. This simple example demonstrates that the factorization error indeed depends on the coupling and can be reduced by updating the coupling.

### 3.3 Rectified Discrete Flow

To enable efficient few-step generation, we propose Rectified Discrete Flow (ReDi), a method designed to directly tackle the underlying factorization error. Based on our finding that this error closely depends on the coupling, ReDi provides a mechanism to rectify the coupling, thereby removing this primary bottleneck to fast and efficient synthesis. Given a coupling $\pi_k$ (initially $\pi_0$),

the rectification process involves training a DFM using $\pi_k$ and subsequently generating sample pairs $(X_0, X_1)$ by sampling from the source distribution and transforming $X_0$ through the trained DFM. These generated pairs collectively define a new coupling, $\pi_{k+1}$ with monotonically decreased factorization error.

Formally, at iteration $k$, the rectification process takes a coupling $\pi_k(X_0, X_1)$ and produces a new coupling $\pi_{k+1}(X_0, X_1)$. This is achieved by first training a DFM, denoted by its conditional distribution $p_\theta(X_1|X_0)$, to model the transition probability defined by the current coupling. Once the DFM is trained, the rectified coupling $\pi_{k+1}$ is formally defined as the joint distribution resulting from sampling $X_0 \sim p(X_0)$ and $X_1 \sim p_\theta(X_1|X_0)$.

$$\pi_{k+1}(X_0, X_1) = p(X_0)p_\theta(X_1|X_0). \tag{6}$$

Then we can show that the rectification process of Eq. 6 is monotonically decreasing the Conditional TC. For brevity, we present an informal statement of this guarantee below. The formal version with assumptions is provided in Appx. A along with its proof.

**Theorem 1** (Informal). *Let $\pi_k(X_0, X_1)$ be a coupling at iteration $k$, and let $\pi_{k+1}(X_0, X_1)$ be the "rectified" coupling obtained via the ReDi procedure at iteration $k$. Then, under certain assumptions, it satisfies the following:*

$$TC_{\pi_{k+1}}(X_1|X_0) \quad \leq \quad TC_{\pi_k}(X_1|X_0).$$

In addition to the theoretical support, we also demonstrated the decrease in factorization error through the rectification process in Sec. 4.4.

Given Thm. 1, we can apply the rectification process iteratively to reduce the factorization error. By repeatedly applying the rectification process for K iterations, starting with the initial coupling $\pi_0$, we obtain a sequence of couplings $\pi_0, \pi_1, \cdots, \pi_K$ with monotonically reduced factorization error. Having small factorization error, this rectified coupling $\pi_K$ is then well-suited for training an efficient one-step generative model, addressing the few-step generation challenge highlighted earlier.

At the same time, the rectification process has to be performed with care: while each step reduces the factorization error (Thm. 1), the marginal approximation error between the target marginal $p(X_1)$ and the rectified marginal $\pi_{k+1}(X_1)$ can accumulate in practice because each $\pi_{k+1}$ is estimated from model-generated pairs rather than real data. This accumulation becomes more pronounced in high-cardinality data (*e.g.*, text), where modeling the target distribution is inherently more challenging. To address this issue, we introduce a perturbed rectification strategy. Instead of using samples from the source distribution $p(X_0)$, the perturbed rectification first sample a clean target $X_1 \sim p(X_1)$ and then perturb it to $X_t \sim p(X_t|X_1)$ at a random time $t$. The rectification is then performed by generating $X_1 \sim p_\theta(X_1|X_t)$. By replacing the source distribution samples with perturbed ones for rectification, we reduce the empirical discrepancy between the true target distribution $p(X_1)$ and the rectified marginal $\pi_k(X_1)$. Detailed algorithm can be found in Appx. C.

ReDi's iterative process closely resembles rectification methods [20, 24, 25, 42] and Iterative Markovian Fitting (IMF) procedures [21, 22, 36]. Similar to Rectified Flow [24], which progressively straightens the transport path between distributions in continuous space, ReDi iteratively refines the coupling to minimize the factorization error, a metric tailored for the discrete domain. Furthermore, the iterative training and data generation steps in ReDi are analogous to the Markovian and reciprocal projections of IMF. These parallels suggest that ReDi can be viewed as an application of the broader principle of iterative coupling refinement, adapted for the discrete domain.

ReDi presents several key advantages that distinguish it from existing distillation methods [10, 17, 31] on discrete data. A primary benefit is its notable simplicity and ease of implementation. ReDi does not introduce specialized objective functions for improving few-step inference, allowing broad applicability to diverse DFMs. Also, unlike some distillation methods that require both teacher and student networks during training, ReDi only handles a single DFM, reducing the training memory requirements. Since the rectification process of ReDi is orthogonal to distillation methods, it can be also applied with the existing approaches to further boost the few-step performance. Finally, we can employ the rectified couplings directly to train one-step generative models. These advantages make ReDi a practical and powerful approach for tackling the few-step generation challenge.

# 4 Experiments

## 4.1 Experimental Setup

**Datasets** We conduct experiments on two benchmark datasets: ImageNet [9] for class-conditional image generation and OpenWebText dataset [14] for text generation. For the image generation task, following prior works [8, 17], we represent 256x256 images as 16x16 vector-quantized tokens using a pretrained VQGAN model [12]. Each token can take one of 1024 possible values. For text generation, following prior works [10, 30, 31], we use sequences of 1024 tokens, tokenized with GPT-2 tokenizer [4] which has a vocabulary size of 50257.

**Baselines** We compared our method with SDTT [10] and Di4C [17], prior works for efficient few-step generation in DFMs. Both SDTT and Di4C distill the teacher model's multi-step behavior into a few-step student model. Di4C suggests its variant, di4c-d, which additionally utilizes a data prediction loss to enhance the performance. For all in our experiments, we used di4c-d as Di4C baseline. As SDTT is tailored for masked diffusion models, we compare SDTT only if the teacher model is masked diffusion model.

**Teacher Models** To compare with distillation methods [10, 17], we utilized publicly available pretrained models as teacher models for the baselines. These models served as the initial models for the first rectification process of ReDi. For ImageNet experiments, we adopt pre-trained MaskGIT as a teacher model [1] based on masked discrete diffusion framework [45]. For OpenWebText experiments, we adopt DUO+DCD [31] as a teacher model that utilizes uniform random state as initial state.

**Implementation** For ReDi, we use the same training objective as the teacher model [8, 31] and finetune the model from the previous iteration. For the one-step distillation model, training uses the same objective while sampling $X_0$ instead of $X_t$ during training, so that the model specifically targeting for direct transition from $t = 0$ to $t = 1$. For image generation, we rectified the coupling with 50k pairs with 16-step heuristic sampling method of MaskGIT [3, 8]. For text generation, we utilized 20k pairs with 1024-step and applied perturbed rectification strategy. For each experiment, we denote $\text{ReDi}^k$ if the model is trained on $\pi_k$. For instance, teacher model in each experiment can be denoted as $\text{ReDi}^0$. In similar, we also denote our one-step distillation model as $\text{ReDi}^k$-distill. The experiments in Sec. 4.4 are conducted on the ImageNet dataset unless specified. Training and implementation details are provided in Appx. B.

**Metrics** For image generation, we generate 50k images and measure Fréchet Inception Distance (FID) [18], Inception Score (IS) [33] following the procedure of prior works [8, 28]. We also measure precision, recall, density, and coverage [27, 32] to further assess the fidelity and diversity of generated images. As DFMs can utilize classifier-free guidance (CFG) [19] to control the generation quality, we report the score with best FID among the CFG values in $\{1, 2, \cdots, 8\}$. We measure the generative perplexity [11] of 1024 texts using the LLaMa 3.1-8B model [15]. To complement the metric, we also assess 1-gram entropy to detect failure modes involving simple, repetitive generation [37].

## 4.2 Results on Image Generation

**Few-step Generation** As shown in Tab. 1 and Fig. 2, ReDi improves 4-step generation performance over its teacher model and achieves comparable results against distillation baselines. Specifically, both $\text{ReDi}^1$ and $\text{ReDi}^2$ outperform MaskGIT's four-step generation, yielding FID scores of 7.58 and 7.86 (vs. 10.90) and Inception Scores of 228 and 240 (vs. 184), respectively. Against other baselines Di4C and SDTT (4-step), ReDi exhibits comparable performance. For instance, while Di4C records a lower FID (6.20), $\text{ReDi}^1$ achieves a higher IS (228 vs. Di4C's 216). Additionally, $\text{ReDi}^1$ maintains comparable scores across Precision, Density, and Coverage metrics.

Fig. 2 further illustrates ReDi's generation quality, which appears on a similar level when compared to distillation baselines [10, 17] while improvement against MaskGIT by reducing the factorization error is clearly shown. To be specific, due to the factorization error, MaskGIT's 4-step generation often leads to artifacts in structures in generated samples. For instance, the goldfish image generated

---

[1]To enable the update of coupling, for ReDi experiments, we finetuned MaskGIT while changing the initial states with stochasticity. Details can be found in Appx. B.1

Table 1: Performance of discrete flow-based models on ImageNet at different generation steps. ReDi$^k$ denotes the model trained on $k$-th rectified coupling $\pi_k$, and ReDi$^k$-distill denotes the model specifically targetting for one-step generation. We reproduced SDTT on ImageNet with 3 round of distillation and denoted as SDTT$^\dagger$.

| Step | Model | FID ($\downarrow$) | IS ($\uparrow$) | Prec. ($\uparrow$) | Rec. ($\uparrow$) | Den. ($\uparrow$) | Cov. ($\uparrow$) |
|---|---|---|---|---|---|---|---|
| 1 | MaskGIT [3] | 95.16 | 12 | 0.26 | 0.12 | 0.17 | 0.35 |
| | SDTT$^\dagger$ [10] | 90.40 | 14 | 0.31 | 0.13 | 0.21 | 0.34 |
| | Di4C [17] | 90.32 | 13 | 0.26 | 0.24 | 0.17 | 0.33 |
| | ReDi$^1$ | 37.43 | 49 | 0.63 | 0.51 | 0.78 | 0.86 |
| | ReDi$^2$ | 21.80 | 90 | 0.74 | **0.52** | 1.05 | 0.93 |
| | ReDi$^3$-distill | **11.68** | **182** | **0.83** | 0.44 | **1.25** | **0.96** |
| 4 | MaskGIT [3] | 10.90 | 184 | 0.83 | 0.46 | 1.18 | 0.96 |
| | SDTT$^\dagger$ [10] | 8.97 | 205 | **0.88** | 0.41 | **1.43** | 0.97 |
| | Di4C [17] | **6.20** | 216 | 0.87 | **0.52** | 1.33 | **0.98** |
| | ReDi$^1$ | 7.58 | 228 | 0.87 | 0.46 | 1.33 | **0.98** |
| | ReDi$^2$ | 7.86 | **240** | 0.87 | 0.44 | 1.31 | 0.97 |
| 8 | MaskGIT [3] | 6.51 | 227 | 0.89 | 0.48 | 1.38 | 0.98 |

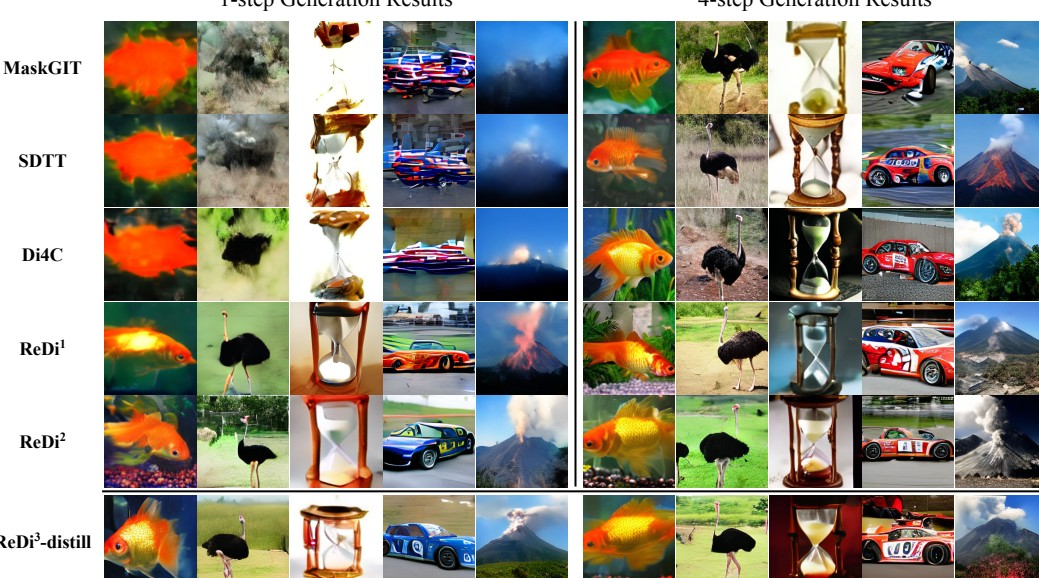

Figure 2: Generated images from various discrete flow-based models. ReDi successfully generate images with natural structures even under one-step generation settings.

by MaskGIT omits the tail of the fish and has heads on both sides. We conjecture that simultaneous sampling of tokens at both sides fails because their joint distribution is poorly factorized when conditioned on the tokens from the previous step. Compared to MaskGIT, the images generated by ReDi shows improved structural coherence. This improvement in structural coherence is attributed to the reduced factorization error achieved by ReDi, allowing the model to better capture inter-dimensional dependencies during generation. Additional qualitative results are shown in Appx. D.5.

**One-step Generation** As shown in Tab. 1, the one-step distillation model trained on the rectified coupling (ReDi$^3$-distill) achieves remarkable performance against 1-step generation of other methods. With an FID of 11.67 and IS of 181, it significantly outperforms their one-step results (e.g., SDTT 1-step FID 90.40, IS 14; Di4C 1-step FID 90.32, IS 13). Interestingly, while the performance of ReDi$^1$ and ReDi$^2$ is inferior to ReDi$^3$-distill, they show significant improvement on one-step generation

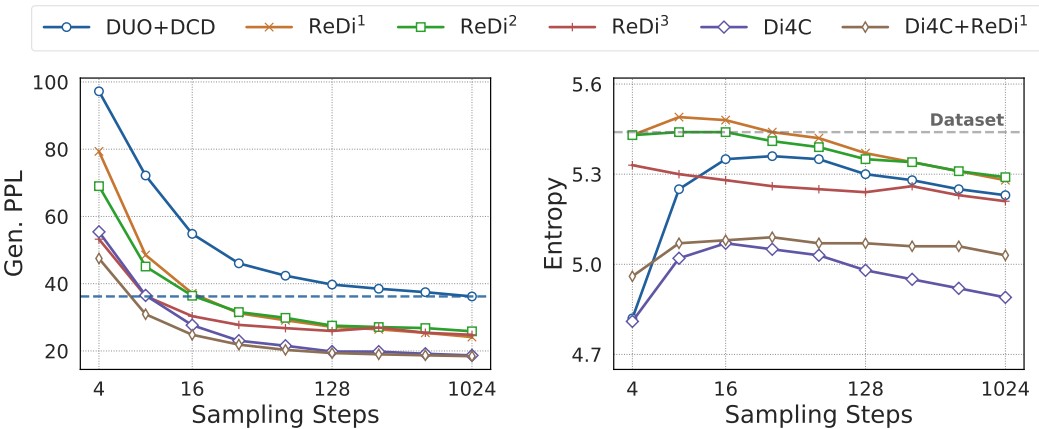

Figure 3: Comparison of various discrete flow-based models on OpenWebText. The blue horizontal line denotes 1024-step generation performance of DUO+DCD. Lower generative perplexity (Gen. PPL) indicates more natural texts. Following Wang et al. [37], we additionally assess the entropy of generated samples to monitor the pitfall of generative perplexity. We provide the exact values for each metric in Appx. D.1

by iteratively applying the proposed rectification process. Compared to our method, SDTT and Di4C fails to improve the performance of one-step generation reasonably. This discrepancy between ReDi against SDTT and Di4C demonstrates that ReDi successfully enables efficient generation in a single step, supports that the rectified coupling is well-suited for one-step generation by reducing the Conditional TC. Furthermore, the performance of our one-step model reaches close to that of the multi-step teacher model (e.g., MaskGIT 4-step FID 10.90, IS 184), highlighting that the better coupling achieved through ReDi's iterative rectification process is highly effective in bridging the gap towards efficient, high-quality one-step generation.

Fig. 2 also visually supports the discrepancy between ReDi and other methods. By iteratively rectifying the coupling, the fidelity of generated images shows significant improvement, while MaskGIT, SDTT, and Di4C fails to generate images in one-step manner. Additional qualitative results are presented in Appx. D.5.

### 4.3 Results on Text Generation

Compared with the image generation, text synthesis is much more challenging due to its significantly larger state space ($1024^{256}$ *vs.* $50257^{1024}$). For text generation, we compare the DUO+DCD which is used as a teacher model, its ReDi-rectified variants (ReDi$^k$), the Di4C-distilled model (Di4C), and a hybrid Di4C+ReDi$^1$. To train the hybrid model, we generate the rectified coupling with Di4C and finetune the Di4C model with the training procedure of DUO [31].

The results are shown in Fig. 3. The blue horizontal line denotes the performance of 1024-step generation with the teacher model. Iterative rectification with ReDi$^k$ (k=1,2,3) consistently yields lower perplexity across few-step settings and boosts the required decoding steps for such quality. Specifically, compared to the teacher model (DUO+DCD)'s 1024-step performance, ReDi$^1$, ReDi$^2$, and ReDi$^3$ show equivalent performance at 16 and 8 steps, achieving significant speedups of x64 and x128. This is consistent with our hypothesis that iterative rectification additionally reduces the factorization error, identified as a main challenge for few-step generation.

To demonstrate ReDi's applicability, we additionally apply it to the distilled model with Di4C, resulting in the combination of ReDi on Di4C (Di4C+ReDi$^1$). Compared to Di4C, applying ReDi (Di4C+ReDi$^1$) enhances few-step performance. These results suggest that ReDi is effective in enhancing few-step generation performance and is applicable across different base models.

To avoid misleadingly low perplexity scores caused by repetitive tokens [37], we also measure the entropy of the generated samples. As shown in Fig. 3 (right), the ReDi variants achieve entropy values

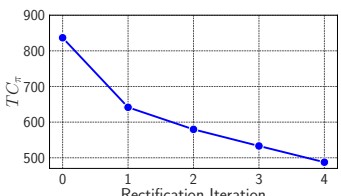
(a) $TC_\pi$ over rectification iteration

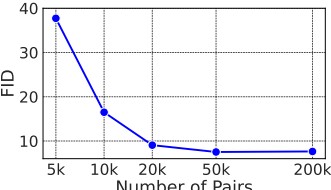
(b) FID over number of pairs

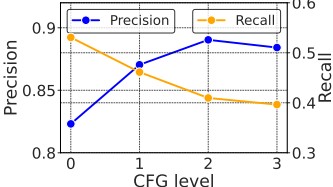
(c) Precision and Recall over CFG

Figure 4: Ablation studies of ReDi on ImageNet. We conducted ablation studies about iterative rectification, number of pairs to represent the coupling, and the effect of decoding strategy.

Table 2: Ablation study on the perturbed rectification strategy. The perturbed rectification strategy reduces the discrepancy between original dataset and rectified coupling.

|              | Data Gen.PPL | Data Entropy | 1024-step Gen.PPL | 1024-step Entropy |
|--------------|--------------|--------------|-------------------|-------------------|
| Orig. Dataset | 9.75        | 5.44         | 36.21             | 5.23              |
| w/ Perturbed  | 26.02       | 5.46         | 24.13             | 5.28              |
| w/o Perturbed | 36.21       | 5.23         | 44.17             | 5.11              |

comparable to the teacher model, indicating ReDi avoids the failure mode. Qualitative inspection of the samples in Appx. D.5 further suggests that ReDi generates plausible text.

### 4.4 Analysis and Ablation Studies

**Empirical Analysis on Conditional TC** To support the theoretical findings in Sec. 3.3, we empirically demonstrate that the proposed rectification process iteratively decreases Conditional TC. To approximate the Conditional TC, we generated 10 samples from the same initial state $x_0$ 5k times (which resulted in 50k samples) and then approximated the Conditional TC based on their frequencies. The progressive decrease of Conditional TC over rectification iterations is shown in Fig. 4a. While Thm. 1 has shown that the Conditional TC is monotonically decreasing, we empirically find that it usually strictly decreases as iterations progress. While the Conditional TC decreases over rectification iterations, we empirically observe that DFM performance degrades in practice due to error propagation arising from sampling pairs from the trained DFM. A similar observation can also be found in the rectification of continuous flows [46].

**Ablation on the Size of Dataset** In the proposed method, the rectified coupling $\pi_k$ is determined by pairs sampled from the distribution defined by a pretrained $\theta$, as shown in Eq. 6. To determine a sufficient number of pairs for reliably estimating the rectified coupling $\pi_1$, we conducted an ablation study on the number of sampled pairs used to compute $\pi_1$ and subsequently trained a DFM using $\pi_1$. The result is shown in Fig. 4b. We reported the 4-step generation performance of the trained model with FID over the number of pairs used to define $\pi_1$. Interestingly, the model's performance starts to saturate with 20k samples and is fully saturated with more than 50k samples. This demonstrates that $\pi_1$ can be effectively defined with a remarkably smaller number of pairs compared to the original dataset, which consists of 1.3M images.

**Ablation on CFG Level for Rectification Pair Sampling** The proposed rectification process relies on $(x_0, x_1)$ pairs sampled from a pretrained DFM $\theta$ to define the coupling $\pi$. To investigate how the DFM's decoding strategy impacts the generation performance of a new DFM subsequently trained, we conduct an ablation study by controlling the Classifier-Free Guidance (CFG) level [19] during sampling pairs. As shown in Fig. 4c, we empirically observe that controlling CFG affects the fidelity-diversity tradeoff of trained DFM. This observation is similar to the observation in rectification of continuous flows [42], despite the inherent differences between discrete and continuous flow mechanisms.

**Ablation on Perturbed Rectification in Text Generation** As discussed in Sec. 3.3, we empirically find that perturbed rectification strategy is required when treating high-dimensional data. The results in Tab. 2 demonstrates that the perturbed rectification strategy narrows the discrepancy between

original dataset and the updated coupling. Without the strategy, the trained model's generative perplexity (44.17) degrades compare to the teacher model's perplexity (36.21) due to the poor quality of training dataset. In contrast, the perturbed strategy enhances the training dataset, leading to the trained model achieving a superior perplexity of 24.13, surpasses the teacher model.

## 5    Limitations and Future Work

While ReDi demonstrates promising results in enabling efficient few-step and one-step generation for discrete flow-based models by rectifying the coupling and reducing factorization error, this work has several limitations that open avenues for future research.

First, the connection between the rectification process in discrete flows (ReDi) and its counterpart in continuous flows (*e.g.*, Rectified Flow [24]) is not yet fully elucidated. While we characterize the error in DFMs using Conditional TC and show its monotonic decrease, a deeper theoretical understanding of the parallels and differences in how rectification improves paths or reduces errors in these two distinct domains could provide further insights and potentially lead to unified frameworks.

Second, our current work primarily focuses on non-autoregressive or parallel decoding DFMs. However, as suggested by recent interpretations (e.g., D3PM [1]), autoregressive (AR) models can also be viewed within the broader DFM framework. Future work could explore extending the ReDi framework to AR models, potentially offering new ways to accelerate their notoriously slow sequential sampling process by rectifying their implicit state transition couplings could be an interesting direction for future research.

## 6    Conclusion

In this paper, we introduced Rectified Discrete Flow (ReDi), a novel iterative method designed to address the challenge of slow, multi-step sampling in Discrete Flow-based Models (DFMs). We identified the factorization approximation, necessary for handling high-dimensional discrete data, as a primary source of error that hinders efficient few-step generation. We rigorously characterized this factorization error using Conditional Total Correlation (TC), highlighting its dependence on the coupling between source and target distributions.

ReDi tackles this issue by iteratively rectifying the coupling. Each ReDi iteration involves training a DFM on the current coupling and then using it to generate a new, rectified set of paired samples, which defines the coupling for the next iteration. We theoretically proved that each rectification step guarantees a monotonically decreasing Conditional TC, ensuring convergence towards a coupling with lower intrinsic factorization error.

Empirically, through experiments on image and text generation benchmarks, we demonstrated that ReDi reduces Conditional TC and improves few-step generation performance. Notably, the rectified couplings obtained via ReDi are particularly well-suited for training highly efficient one-step generative models, achieving remarkable performance that often surpasses or rivals more complex distillation techniques. ReDi offers a simple, theoretically grounded, and broadly applicable approach that avoids specialized training objectives or teacher-student architectures typically required by existing DFM distillation methods.

By providing a new perspective on improving discrete data synthesis through direct coupling manipulation, ReDi advances the development of faster and more efficient generative models applicable to a wide range of discrete data modalities.

**Acknowledgments**    This work was in part supported by the National Research Foundation of Korea (RS-2024-00351212 and RS-2024-00436165), the Institute of Information & communications Technology Planning & Evaluation (IITP) (RS-2022-II220926, RS-2024-00509279, RS-2021-II212068, RS-2022-II220959, and RS-2019-II190075), and the "HPC support" project funded by the Korea government (MSIT), and the Korea Meteorological Administration Research and Development Program "Developing Service Platform Technology for AI and Data Convergence" (KMA2021-00122).

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

# Appendix

## A    Theoretical Analysis

### A.1    Preliminaries

We first begin by introducing the definition, assumptions, and the property of KL divergence for our formal proof.

**Definition 1** ($M$-step Decoding Process). *With a discrete flow-based model $p_\theta$, $M$-step decoding process is defined as:*

$$p_\theta(X_1|X_0) = \sum_{X_{t_1},\dots X_{t_{M-1}}} \prod_k \prod_i p_{\theta,t_{k+1}|t_k}(X^i_{t_{k+1}}|X_{t_k})$$

**Assumption 1.** *Let $P$ be the family of M-step decoding processes. We assume that our model $p_\theta(X_1|X_0)$ lies within the log-convex hull of $P$.*

This assumption is justified because for a sufficiently large number of steps $M$, the family of processes $P$ can closely approximate the hypothesis space of $p_\theta(X_1|X_0)$.

**Assumption 2.** *We assume that at each rectification step, the model $p_\theta$ is the minimizer of the objective function, $D_{KL}(p_{\pi_k,1|0}(X_1|X_0)||p_\theta(X_1|X_0))$.*

**Property 1** (Pythagorean Inequality for KL Divergence [40]). *For a distribution $q$ in a log-convex set of distributions $Q$, if $q^* = \arg\min_{q \in Q} D_{KL}(p||q)$ and $r \in Q$, then*

$$D_{KL}(p||r) \geq D_{KL}(p||q^*) + D_{KL}(q^*||r).$$

### A.2    Formal Theorem and Proof

**Theorem 1.** *Let $\pi_k(X_0, X_1)$ be a coupling at iteration $k$, and let $\pi_{k+1}(X_0, X_1)$ be the "rectified" coupling obtained via the ReDi procedure at iteration $k$. Then, under Assumptions 1 and 2, the following inequality holds:*

$$TC_{\pi_{k+1}}(X_1|X_0) \quad \leq \quad TC_{\pi_k}(X_1|X_0).$$

*Proof.* The proof proceeds as follows:

$$
\begin{aligned}
TC_{\pi_k}(X_1|X_0) &= \mathbb{E}_{X_0}[D_{KL}(p_{\pi_k,1|0}(X_1|X_0)|| \prod_i p_{\pi_k,1|0}(X^i_1|X_0))] \\
&\geq \mathbb{E}_{X_0}[D_{KL}(p_{\pi_k,1|0}(X_1|X_0)||p_\theta(X_1|X_0))] \\
&\quad + \mathbb{E}_{X_0}[D_{KL}(p_\theta(X_1|X_0)|| \prod_i p_{\pi_k,1|0}(X^i_1|X_0))] \\
&\geq \mathbb{E}_{X_0}[D_{KL}(p_\theta(X_1|X_0)|| \prod_i p_{\pi_k,1|0}(X^i_1|X_0))] \\
&= \mathbb{E}_{X_0}[D_{KL}(p_{\pi_{k+1},1|0}(X_1|X_0)|| \prod_i p_{\pi_k,1|0}(X^i_1|X_0))] \\
&= \mathbb{E}_{X_0}[D_{KL}(p_{\pi_{k+1},1|0}(X_1|X_0)|| \prod_i p_{\pi_{k+1},1|0}(X^i_1|X_0)) \\
&\quad + \sum_i D_{KL}(p_{\pi_{k+1},1|0}(X^i_1|X_0)||p_{\pi_k,1|0}(X^i_1|X_0))] \\
&\geq \mathbb{E}_{X_0}[D_{KL}(p_{\pi_{k+1},1|0}(X_1|X_0)|| \prod_i p_{\pi_{k+1},1|0}(X^i_1|X_0))] \\
&= TC_{\pi_{k+1}}(X_1|X_0). \quad \square
\end{aligned}
$$

(7)

(8)

The inequality in Eq. 7 follows from Property 1, which is applicable under Assumptions 1 and 2, while the equality in Eq. 8 follows from the definition of rectification process in Eq. 6.

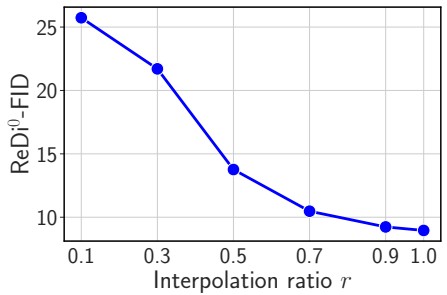

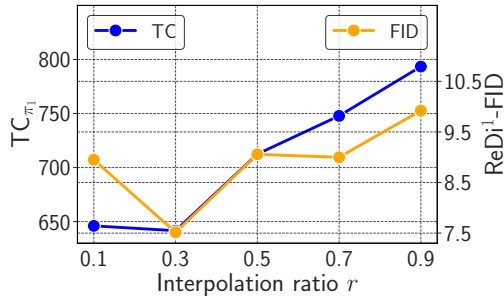

(a) 16-step FID over interpolation ratio $r$      (b) $TC_\pi$ and 4-step FID over absorbing ratio $r$

Figure 5: Ablation studies about finetuning MaskGIT with stochastic initial states.

# B Implementation Details

This section outlines key implementation details, covering the procedure for finetuning the masked state diffusion model used in Sec. 4.2 for rectification, and the hyperparameters for both generating the rectified coupling and training ReDi models used in Sec. 4.

## B.1 Finetuning MaskGIT with Stochastic Initial States

While Thm. 1 holds for any coupling $\pi_k$ between a source $X_0$ and a target $X_1$, the equality condition is met if the source is a masked state $\mathbf{m}$ (i.e., $p(X_0) = \delta_{\mathbf{m}}(X_0)$). This occurs because $\pi(X_0, X_1) = p(X_1|X_0)p(X_0) = p(X_1)$ for any $p(X_1|X_0)$ preserving the target distribution. Therefore, to handle models with a masked initial state [1, 3, 8], we finetune the models by modifying the initial distribution $p(X_0)$ as:

$$p(X_0) = (1 - r)u(X_0) + r\delta_{\mathbf{m}}(X_0), \tag{9}$$

where $u(X_0)$ is the uniform distribution over possible states, and $r \in [0, 1]$ is a interpolation ratio interpolating between the delta distribution $\delta_{\mathbf{m}}(X_0)$ and the uniform distribution $u(X_0)$.

To analyze the effect of the interpolation ratio $r$, we finetune MaskGIT models with $r \in \{0.1, 0.3, 0.5, 0.7, 0.9\}$, which we denote as ReDi$^0$. After training, we evaluate the models by 16-step generation FID using a CFG level of 1. We also measure the Conditional TC of the rectified coupling $\pi_1$ and the 4-step FID of the corresponding models, ReDi$^1$. The original MaskGIT model [3] corresponds to $r = 1.0$ in this setup. As shown in Fig. 5a, decreasing $r$ degrades the performance (FID) of ReDi$^0$. We conjecture the degradation is due to the distributional mismatch between the initial state of the original MaskGIT ($r = 1.0$) and that of finetuned ReDi$^0$ models (which use $r < 1.0$). Although the performance of ReDi$^0$ degrades with smaller $r$, we observe that reducing $r$ also decreases the Conditional TC of $\pi_1$ (Fig. 5b). Consequently, a trade-off emerges between ReDi$^0$-FID and the Conditional TC of $\pi_1$. This balance yields the optimal ReDi$^1$ model performance (ReDi$^1$-FID) at $r = 0.3$. Accordingly, we use $r = 0.3$ for all image generation experiments in the main paper.

## B.2 Hyperparameters

**Hyperparameters for Image Generation**    For finetuning MaskGIT [3, 8] with stochastic initial states, we use the AdamW optimizer with a learning rate of 1e-4 and a weight decay of 1e-5. The model is trained for 13 epochs on the original ImageNet dataset [9], with each epoch consisting of 1.3M images. The global batch size is set to 512, and the training take 96 GPU hours with A6000 GPUs.

For training the ReDi variants, we set the global batch size of 512 and applied a cosine learning rate scheduler with decay over 100 epochs. Each epoch consisted of 50k pairs. To rectify the coupling, we control the decoding parameters during pair generation. For ReDi$^1$ and ReDi$^2$, pairs for the coupling were generated with a CFG level of 1 and 16-step decoding. For the ReDi$^3$-distill model, pairs were generated with a CFG level of 8 and 16-step decoding. Each of ReDi variants take 16 GPU hours with A6000 GPUs.

**Hyperparameters for Text Generation**  For all ReDi variants, we use a learning rate of 3e-4. The models are trained with a linear warmup over the first 2500 steps. The global batch size is set to 128. For Di4C+ReDi[1], we fixed the random variable $\lambda$ for the control variates in Di4C [17] as $\mathbf{0}$. The best-performing model is selected and used after completing 25,000 training steps. Each training takes 12 GPU hours with H100 GPUs.

## C  Standard and Perturbed Rectification

---

**Algorithm 1** Coupling Update (Standard)

---

1: **Input:** $p(X_0)$, $p_\theta$, dataset size $N$
2: **for** $i = 1$ **to** $N$ **do**
3:     $x_0^{(i)} \sim p(x_0)$
4:     $x_1^{(i)} \sim p_\theta(x_1|x_0^{(i)})$
5: **end for**
6: **Return:** $\{(x_0^{(i)}, x_1^{(i)})\}_{i=1}^N$

---

**Algorithm 3** Training (Standard)

---

1: **Input:** $\{(x_0^{(i)}, x_1^{(i)})\}_{i=1}^N$
2: **while** converge **do**
3:     Sample $(x_0, x_1)$ from $\{(x_0^{(i)}, x_1^{(i)})\}_{i=1}^N$
4:     $t \sim \text{Uniform}(0, 1)$
5:     $x_t \sim p(x_t|x_0, x_1)$
6:     $\mathcal{L} = \text{CrossEntropy}(p_\theta(x_1|x_t), x_1)$
7:     Backpropagte and update parameters
8: **end while**

---

**Algorithm 2** Coupling Update (Perturbed)

---

1: **Input:** $p(x_1)$, $p_\theta$, dataset size $N$
2: **for** $i = 1$ **to** $N$ **do**
3:     $x_1^{(i)} \sim p(x_1)$
4:     $t \sim \text{Uniform}(0, 1)$
5:     $x_t^{(i)} \sim p(x_t|x_1^{(i)})$
6:     $x_1^{(i)} \sim p_\theta(x_1|x_t^{(i)})$
7: **end for**
8: **Return:** $\{(x_t^{(i)}, x_1^{(i)})\}_{i=1}^N$

---

**Algorithm 4** Training (Perturbed)

---

1: **Input:** $\{(x_t^{(i)}, x_1^{(i)})\}_{i=1}^N$
2: **while** converge **do**
3:     Sample $(x_t, x_1)$ from $\{(x_t^{(i)}, x_1^{(i)})\}_{i=1}^N$
4:     $\mathcal{L} = \text{CrossEntropy}(p_\theta(x_1|x_0), x_1)$
5:     Backpropagte and update parameters
6: **end while**

---

Algo. 1- 4 highlight the key difference between standard and perturbed rectification. The standard rectification updates the coupling of $(X_0, X_1)$ and samples $X_t$ from $p(X_t|X_0, X_1)$ during training. In contrast, the perturbed rectification updates the coupling of $(X_t, X_1)$ and use $X_t$ directly during training. As the time $t$ is randomly sampled from $[0, 1)$, the perturbed approach also covers the standard rectification case at $t = 0$.

## D  Additional Results

### D.1  Detailed Values for Fig. 3

We provide the detailed values for each metric that corresponds to Fig. 3 in Tab. 3, 4 for future research.

Table 3: OpenWebText generative perplexity scores.

|      | DUO+DCD | ReDi[1] | ReDi[2] | ReDi[3] | Di4C  | Di4C+ReDi[1] |
|------|---------|---------|---------|---------|-------|--------------|
| 4    | 97.22   | 79.35   | 69.01   | 53.24   | 55.42 | 47.50        |
| 8    | 72.18   | 48.53   | 45.11   | 36.33   | 36.52 | 30.92        |
| 16   | 54.82   | 37.13   | 36.42   | 30.34   | 27.66 | 24.81        |
| 32   | 46.05   | 31.21   | 31.56   | 27.75   | 23.04 | 21.88        |
| 64   | 42.38   | 29.12   | 29.85   | 26.78   | 21.54 | 20.32        |
| 128  | 39.74   | 27.16   | 27.53   | 25.93   | 19.82 | 19.38        |
| 256  | 38.50   | 26.42   | 27.11   | 26.96   | 19.79 | 19.00        |
| 512  | 37.48   | 25.38   | 26.80   | 25.41   | 19.16 | 18.70        |
| 1024 | 36.21   | 24.13   | 25.84   | 24.80   | 18.68 | 18.44        |

Table 4: OpenWebText entropy scores.

|  | DUO+DCD | ReDi[1] | ReDi[2] | ReDi[3] | Di4C | Di4C+ReDi[1] |
|---|---|---|---|---|---|---|
| 4 | 4.82 | 5.43 | 5.43 | 5.33 | 4.81 | 4.96 |
| 8 | 5.25 | 5.49 | 5.44 | 5.3 | 5.02 | 5.07 |
| 16 | 5.35 | 5.48 | 5.44 | 5.28 | 5.07 | 5.08 |
| 32 | 5.36 | 5.44 | 5.41 | 5.26 | 5.05 | 5.09 |
| 64 | 5.35 | 5.42 | 5.39 | 5.25 | 5.03 | 5.07 |
| 128 | 5.30 | 5.37 | 5.35 | 5.24 | 4.98 | 5.07 |
| 256 | 5.28 | 5.34 | 5.34 | 5.26 | 4.95 | 5.06 |
| 512 | 5.25 | 5.31 | 5.31 | 5.23 | 4.92 | 5.06 |
| 1024 | 5.23 | 5.28 | 5.29 | 5.21 | 4.89 | 5.03 |

## D.2 Training Cost Analysis on ImageNet

Table 5: Training cost comparsion on ImageNet.

|  | Dist. Iteration | GPU Hour / Iter. | Total Training Time |
|---|---|---|---|
| MaskGIT [3] | 0 | 1800h | 1800h |
| SDTT[†] [10] | 3 | 68h | 204h |
| Di4C [17] | 1 | 50h | 50h |
| ReDi[2] | 2 | 15h | 30h |
| ReDi[3]-distill | 3 | 15h | 45h |

We further analyze the efficiency of ReDi by measuring GPU hours per iteration with A6000 GPUs. We include the costs to generate the rectified couplings for ReDi variants. As shown in Tab. 5, ReDi demonstrates its training efficiency in both per iteration and total cost. This efficiency is achieved by following reasons. First, as discussed in Section 4.4. and Figure 4(b) in the main paper, the rectification process can be trained with only a small portion of the entire training data (50K images vs. 1M full training data). Therefore, it greatly reduces the cost of forwarding pre-trained models and enhances convergence speed of each rectification iteration. Second, unlike other distillation approaches, ReDi requires only the student model during training, avoiding the cost of operating two models simultaneously.

## D.3 Rectification CFG Ablation on ImageNet

Table 6: ReDi[1] rectification CFG ablation.

| ReDi[1] (4-step) | FID | IS |
|---|---|---|
| CFG 0 | 9.76 | 163 |
| CFG 1 | **7.52** | 228 |
| CFG 2 | 7.77 | 252 |
| CFG 3 | 8.63 | **283** |

Table 7: ReDi[3]-distill rectification CFG ablation.

| ReDi[3]-distill (1-step) | FID | IS |
|---|---|---|
| CFG 1 | 14.11 | 139 |
| CFG 2 | 13.25 | 150 |
| CFG 8 | **11.68** | **182** |

We ablated the CFG level that is used during rectification. As shown in Tab. 6, 7, we found that the optimal CFG value required for rectification and distillation differ. We used the best CFG values for rectification in our main experiments.

## D.4 Inference CFG Ablation on ImageNet

We ablated the CFG level at inference time and reported in Tab. 8, 9. Similar to other discrete flow-based models [8, 17, 34, 39], the performance of ReDi varies with different CFG values.

Table 8: ReDi[1] inference CFG ablation.

| CFG | 1 | 2 | 3 | 4 | 5 | 6 | 7 | 8 |
|---|---|---|---|---|---|---|---|---|
| 1-step | 55.35 | 39.33 | **37.43** | 39.62 | 42.46 | 45.27 | 47.75 | 49.90 |
| 4-step | 22.39 | 12.49 | 9.22 | 8.06 | 7.69 | **7.58** | 7.59 | 7.62 |

Table 9: ReDi[2] inference CFG ablation.

| CFG | 1 | 2 | 3 | 4 | 5 | 6 | 7 | 8 |
|---|---|---|---|---|---|---|---|---|
| 1-step | 26.59 | **21.80** | 22.30 | 23.38 | 24.54 | 25.51 | 26.33 | 27.22 |
| 4-step | 11.47 | 8.68 | 8.03 | **7.86** | 7.86 | 7.90 | 7.92 | 7.95 |

## D.5 Additional Qualitative Results

In addition to Fig. 2 in the main paper, we additionally visualize the 1-step and 4-step generation in Fig. 6 and Fig. 7. As discussed in Sec. 4.1, one-step generation result of ReDi[3]-distill shows comparable fidelity against 4-step generation result of MaskGIT [3], and ReDi models show comparable 4-step generation quality against SDTT [10] and Di4C [17].

We also provide qualitative results for text generation in Fig. 8–13. As discussed in Sec. 4.3, ReDi generates plausible texts. We use the `unidecode` package for decoding Unicode characters.

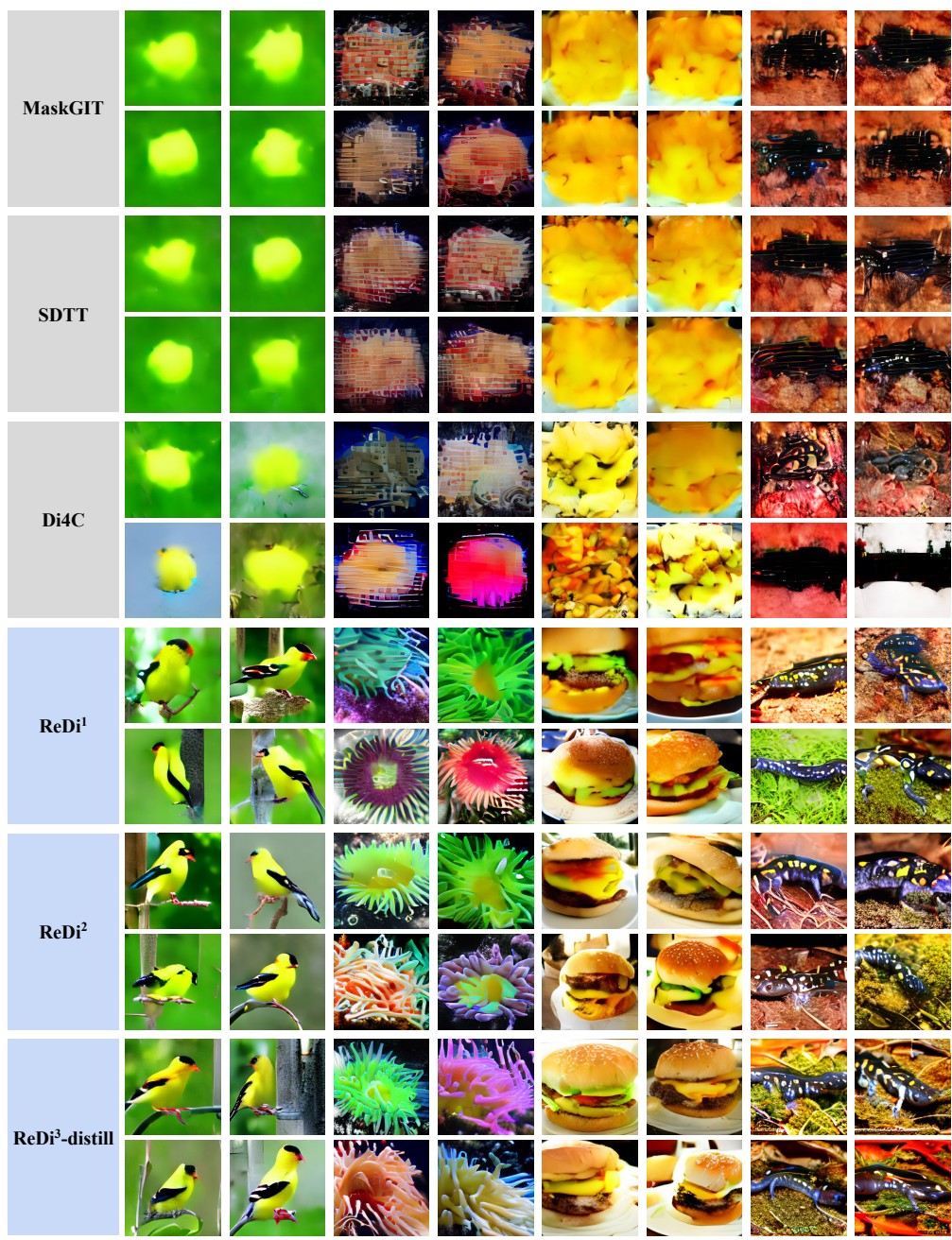

Figure 6: 1-step generation results on ImageNet. Visualized class labels: 11, 108, 933, 28

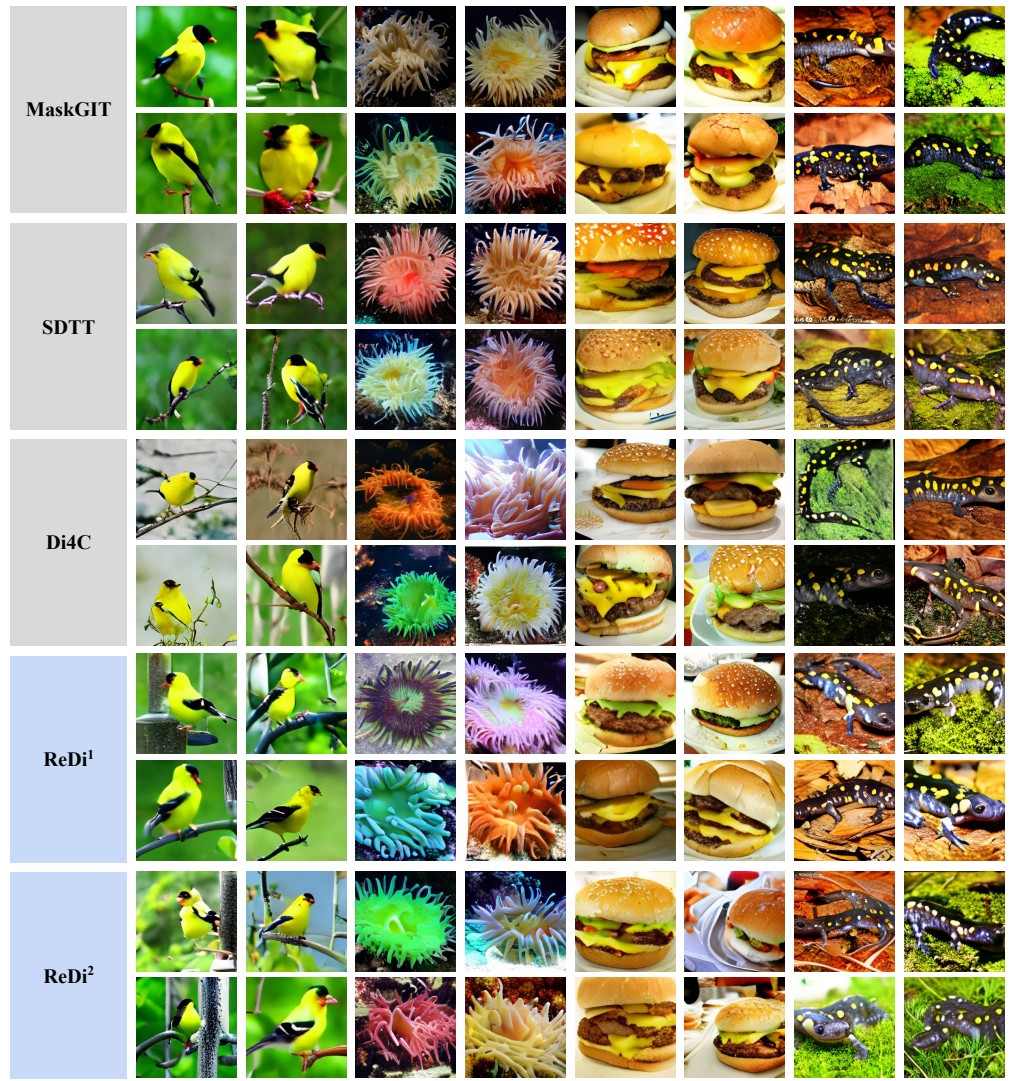

Figure 7: 4-step generation results on ImageNet. Visualized class labels: 11, 108, 933, 28

<|endoftext|>. Recently, my older sister showed up in Seoul,, she was shown some of the monuments and and and buildings. I was a 12 yearyear-old girl, and she was told that these monuments have be be damage<|endoftext|> I would definitely like to make any thoughts on these monuments, as in my experience these are put in the plans, and so it very impossible to ensure they are be removed, and not, they not part of the new level. However, however, even though the project took place in the major capital city, and one of the recent studies found in it has reduced its entire carbon footprint, in the years since he took office. On the other hand, if you don't act this, there will be more more and Also, it has changed so much since you was there, many lots of construction and new state of the art high-rises.

He has special interests in urban design and the environment.<|endoftext|>Dear Reader, As you can imagine, more people are reading The Jerusalem Post than ever before. Nevertheless, traditional business models are no longer sustainable and high-quality publications, like ours, are being forced to look for new ways to keep going. Unlike many other news organizations, we have not put up a paywall. We want to keep our journalism open and accessible and be able to keep providing you with news and analysis from the frontlines of Israel, the Middle East and the Jewish World.

St. Louis, that Paris and Paris signed plans for the settlement. Beijing said the is is isancing progress on building settlement of the Jewish State and the claimsand there will be more support. the city is working up a new percent000-free "itary buffer wall" in Jerusalem late last week, an Paris city official said.

Laredo, and Tel Aviv.

Israel announced development for economic and economic development of the construction,, it was contacted by Ma'a.. the Gaza Authority that it the going to to., Avi Meirman, head of thes UN agency in Gaza Israel the Israel and international other have in the and and the aid and to focused on the development in Gaza's local sector.

End's note on May is a was written by The.<|endoftext|>EMBED | Portland, Ore. If his memory doesn't translate well, a fifth grad is diagnosed with Alzheimer's dementia - is the target of a local high student.

The root project, which was placing a log, was that he wanted to do, to of of photographic memory, as well. It's enough for him.

But the fourth grad, who died Tuesday, had own own needed worst moment. When Paul McCallum in the classmates at a, of summer summer camp to keep him from having a memory-like memory, it was figured, who was placing the place the theighthouse, and the event to take place.

It to everyone at the camp.

"I said 'He's gone.' I't tell him to go," said said at the camp.One of the other people, me said, 'Where's me?' I want him to see it. I've really graduated as a student, I've even done my my yet.

See more the let's thoughts.

Ben McDermott

Milwaukee, Wis, in a good state of his young mind and self-consciousjection, with the a 4-2 record (and 3.2 scoring record) on the team, he a senior year senior.

He is one of the best players of college young career, an incredible talent that, to making a number of plays in game-day, practices and games. Consistently. Looking the him, he has, crunching, outages, doing doing very good job completing it. Taking over the bulk of the team, with a very good things, doing is doing is job.

More info: Wisconsin's transition from due diligence to practice

Speaking to the media, and said he is his to on in offense, and he's perfect fit, and the president of the team that he was going to step down the they thought of him in the NBA draft too Ben O'an wants to break out with the team,, he he is starting center in the team, and he's a kind of starter of quarterback. But isn't the not of hope in sight now.

"He is not the smartest intelligent. Not that the smartest kid but but but the smartest kid," an insider told the Sentinel.

"He went through a period last year where he was slow, and he lost his high sense of alertness. Really," Ron source said, and with the "cuteure to be done." After the<|endoftext|>

Figure 8: A text sample generated by 8-step decoding with DUO+DCD [31].

<|endoftext|> and fift to in the country of sweeping the country, and he says, "What I am very passionate about is trying to spread this country around the world, and this is because I have spent a lot of years and years, in the process of studying the character and growth and character of this country, of the history of it. These discoveries are published in many of the books, books and magazines around the world, and also in the history of its history and development, in books written by the people of the Netherlands, the Netherlands, Germany, and, and now, in the United States."

Holland he says, at the moment he read the book, "One of the greatest people in the world, who know who who lived in the world, and who shaped the character of the country, the modern country, it's one of the most powerful books in the world, and it has attracted a great number of people to America. It's one of the most important books that I read written the America.
"

It is the perfect book to look at the history and the character of the country, of America," he says, in a "dark very different way," a difficult place. "It is the most important book of this time, and to some extent, the book is the willingness of the American people to the struggle to find their place in the history and development of the development of this country all over the world. For the young people, there has been a lot of the power and potential of the country, in the world, to the world, if you are putting your place in America and America."

Advertisement

Holl, the African American in the book,, in the U.S., has spent so much of the rest of his life sitting in the chair hall of the universities, in thinking of, and helping the students in get their degrees, listening to their ideas, thinking about the character, all these countries, and studying them, and some of the basic ideas of how a young man ought better succeed in the universities, in the United States, in Europe, and around the world in the education, teaching, the education and the work of young people.

"But in my youth, the people of the Europe, which is now one of the most powerful countries, has come to be European and European," he says, are pushed to the extreme right of the American political system and start to think in some way, and George he once said, "and for a time we didn't know the truth of what is was going to happen."

In fact, George Bush first became elected president of the United States, Jan June 8, in 1968.

Advertisement

Holland, a professor who now at the University of London,, when he first reading the book, that he read him with being a "great man, a good citizen of this country, of this great and great country."

He says, however, "From the very beginning, Bush's life he was an ordinary man, and he followed the history of the current world in Eastern Europe and Europe, and what it mean if Europe didn't no longer dominate the world. He was a young man, and he followed the course of Europe, and the development of the modern world."

He has spent almost his entire career writing, he says, "Bush's writing said that, and that, the European Union was great, and, the success of a lot of people around the world. It was because of other people's success, that he became an great citizen. From Paris to Paris, he came to the United States.

Advertisement

"From there, he, with the help of the people of the continent and around the world, realized that for many people, America was a powerful country. The power of these people, people from other countries and Europe, and people from the U.S. together, these people became an important part of an American."

Brown, the only American in the book in the United States is in the Netherlands. Last year, in Brown family lost Daniel and James, as a boy, in the streets of the street, the horrors of living and traversing their country with the world, real and unseen, everywhere, and everything that surrounds them and haunts them.

Advertisement

Last year, he says, in Baltimore, the heart of a still-universe Baltimore, that was the beacon of hope. "It was all about our life, and when we got on the plane to the United States, just a few years ago, and this thing happened, and, it had come to the whole States, in the U.S. and became one of the most important stories of the time<|endoftext|>

Figure 9: A text sample generated by 8-step decoding with Di4C [17].

<|endoftext|> follow a meteorite-Earth-like axis of motion, even as the Sun away from the Turbine hit the Earth.

The planet nucleus and some other objects are puzzled by the shift and flatterings, according to the recent paper "The Population of Comet Objects," because objects between bodies have interplanetary mass. For example, some matter belonging of smaller objects reside in the planet the nucleus, in groups with masses of up to 400 times more mass in common.

The core of the earth life-giving has come into contact with a host of other celestial bodies, one big one from Earth. Every year since, increasing numbers of people, astronomers and scientists have steadily expanding the search for the universe, turning from theories and hypotheses, the theory that it would be "diff easy" in 1,000 ago on the basis of evidence that we have a few billion yet to see it.

Image:

a. (c) Gravity and Space, Inc / Mark Tinglin

Mostories alike remain cautious. Details remain as to how big or massive will become. And yet there is one place where a lid keeps a lid on the rate ofcheting-up.

We don't have any good answers about what we would do after the extinction took hold, but we don't think the extinction is still filling the void left by an asteroid, which swept the world east of the sun in 2000. March 23, 2001, recorded on Terra, saw that record low and cold temperatures, producing a series of dust clouds of low pressure that caused a severe and likely extinction. Life at that was more than 250 million cubic kilometres ( cubic miles) in total. At this time, it seems, the broadening of the Sun's away from the core of our core, which transgresses the basic tenets of Islamic faith, threatens our planet and surrounding regions as well

According to a report released by Saudi Arabia's Health Ministry on April 3, new cases of infection are reported in around 1,100 cases each day under the new Saudi kingdom's sharia law, designed to restrict the supply of medicine in the Kingdom of Arabia.

Those with long histories of Islam will note that some leaders of Riyadh fundamentalist Salafafist movement in the government of Saudi Arabia, have tried to lead the movement underground by receiving millions of dollars in cash from Iran and al-Qaeda of the Muslim Brotherhood (AQAP), among other things. But betters, things gets better unless these clerics stand up and take action.

Read more about today's reenlisting

Subscribe to the Guardian 's English-style news email every Monday 31 December

In an interview, Dr. Juan Martinez Rubens, the Geneva-based regional director, told the Financial Times in Geneva. "A human cell is spinning ahead of gravity, in the sense of the cell's rotating mass, so that the incontradable processes are manageable and you can get the good results. This highlights the fact that in the body's evolution through evolution, something new material either accumulate in the body else it undergoes damage by the elements–the same thing goes for the brain. The shape of the limbs is affected these changes," he said.<|endoftext|>For those who know the thermoklosan (EGS) compound found in many plants is responsible for the growth and destruction of many proteins and organ systems (Li and Wobler, 2003). ERS proteins are responsible for the surface proteins and have been characterized by attractive "dynamic dynamics" mixing of dissolved biomolecules on surfaces. Egel is similar, affectionately being referred to as "all else's such particular lipids and hemispheres."

Egel calls these mechanisms an "illustance chemistry" with strong mixing behavior and fundamental electron rotation. What Conrad et al. and Bayes In-Seq Carbon-Claudine vol. (Figure 5; Herschel et al. 1999) call the 'static' effects of heating and 'interration' is the oxidization to which the floor's lipoprotein H and metallophosphate [pH2O2] at room temperature swells with dissolved organic compounds as far away as the floor's solenvironments."

Today, room temperature is generally reported to drop to 10-20 degC (69.5/C) in preference to 40 degC F. Not surprising has alteration of the oxygen-fogal state over centuries taken a such a global view of all organisms. For example, MEAs are damaged by CO2 excretion in Type 1 and Type 2A; Bartletta, 2005. Whereas oxidative damage in Molybic-sensitive MEAs has increased for many decades (Durkin et al., 2005), it is a relatively global view of all changes in the environment. Indeed, in this respect it is a challenge to understand molecular processes in order to explain the<|endoftext|>

Figure 10: A text sample generated by 8-step decoding with ReDi[1]

<|endoftext|> home in Highland Park in August 2007.

Police said Frank sat down with some employees of Buickler Industries who said they found several items that they said had managed to disappear from behind.

"Several items in the yard had gone missing including a suit, wire harness and a chair, one thing missing," police said. "The other was missing as well, a found rug, a skull, jewelry worn by a man a schnapeer had gone missing and all that he was doing had gone missing."

As soon as Frankley came home, employees quickly tracked down the man and discovered many items that he owned until he was at a building insurance company, police said.

To his surprise, the debris was left there for days but which is when the company asked an appraiser for help an explanation.

He said he asked "many questions" and received expert feedback, including the project supervisor who did excavation and tests.

"Their report is so much better than the situation" said Lockhart. "CHASE, "Part I and Part II: A Criminal Investigation,"" reads the official report.

But the project foreman contacted a real rep who for months negotiate the appraiser's ownership of the yard as a dumping ground months later. She and the Buildersers Basingers eventually agreed to include Frank's name into the report.

When Frank Curley, who worked as a high school teacher with three managers moved to Chicago in 2007 and flew to the airport, Frank was found abandoned with stolen stolen toys all while trying to find a new home on his feet.

He's leading a crime task force and said he reportedly found himself caught up in crime.

"There's a two-word movement linking him to organized crime," says Lockhart.

CNN went to visit his home and spoke with reporters. Van Huey, where Frank lives is seen watching helplessly with his daughter on TV.

His family photo still shows Frank's home. (Photo: FASHION HAYY)

"He sure dressed things up with that ace on on screen H Huey tells FOX News.

Although Frank has been identified by police, there have been suggesting he has been used to appear on Facebook, YouTube and more.<|endoftext|>Ital light is very dark in the environment but when deerred in in a room, often it is not visible, especially on the console version of Destiny.

Jiteun says the hypothesis is crucial to make sure players work in conditions that avoid acidity and extreme stress.

"The idea of the light is IR," he told Ars. "This really is all about infrared exposure. It makes it really important to just have it in there that you can paint a picture."

With the hypothesis accepted, the development team says they're moved one step closer to working with Hough, Nell and Guillamics, and they's working in both California, Australia and Japan.

"It's an enormous enthronement that gives very a lot of energy to important exploration and action," Chun said. "This holds that the world can be shaped by our ability to get better images of ourselves in order to do other things than human contact."

In the console version of the game, players can take you around a cave-like world with a set of environmental stables crafted with a cast of material. You duld the object with various materials to tear down or one of the objects sets on fire for health or sets another to mark its crumbs.

It demo demoed by Tom Hough, producer Andrew Nell on Sony Sony Blu-ray 6.0 and PS4 Compact. PC version is available here.

We'd have a program in which it was really compelling to see how the mission would work," Jiteun said. "Where we've had really good success is with some of the Caribbean's blue white ships rolling in the ocean off the coast.

"The actual blue ships rolling around are different from the environments of the game and the footage."

If you choose to use pure blue or not really blue, it should be real instead of dark black, the team says.

"If the environment is completely red, the sky should be pure yellow instead of dark black; the blue should be visible," Jiteun said. "If you can't see below the surface, you just put in some dark black. You can't see it and then you can see it because it's completely there."<|endoftext|>"How it could be more convenient to have healthy air than buy books online?"Dominic Scherbacher said.

The German striker added and new technologies such as haze could lead reduce the damaging effects of heat and pollution. The landmark study, published today by an international team found more than 77,000 people spent PS86m as part of the air<|endoftext|>

Figure 11: A text sample generated by 8-step decoding with ReDi[2]

<|endoftext|> in search those in need

Copyright by WKRN - All rights reserved Mike Miller - In Nashville's charity-raising reported Larry Wright received a phone call that told people the city is paying as little $4 an hour for 36 hours and free to alleviate their needs on the streets is just what a truck can get there. (4:30 Uber and MAX)

Another man was killed in his life, but a growing concern is growing as the situation ramped up Wednesday morning when a rescue worker was driving his Countachio Cavalry. Baggi, 43, recently owned an Italian home and began work in Central Italy. He hopes the relief money will help him make some money and in addition to a few of road trips he makes this year

B Baggi said he'd like to receive supplies as order to anticipate food supplies and damage.

"I don't feel the need to drive any more. I don't get drinking drinking water," he said.

Even if the man doesn't want to get some precious food after finding firewood, he's talking about driving cars without running water, "without shelling" out of his car.<|endoftext|>In the concern that the crisis is likely to caused more suffering in Europe, four top international aid groups from the University of California, Santa Barbara have held Mass at schools in Washington. On Tuesday, the groups with officials from Belgium and Kuwait at the G2020, the G20 economic summit for the that-wealthrich nations.

The United States government and foreign administration have also to normalize the current post-The Security and Climate Council accord between the member states and the EU member states to bring peace to Europe. The negotiations take place in 50 states, with roughly 8,000 men, 700 and 700 countries and governments. U.S. Rep. Bill E. Lee, a member of the accord accord, threatens to cost more than 3,000 lives and has become the U.S. target of the humanitarian crisis.

Davies of the Army called the "Warrior" led the efforts of states affected by war in World War II under the Initiative of 1789, and a total of 44 states led one of the 672 Rep. Lee forces to the position they had fallen. The majority of the victims are women. At least 39 children guarding the victims are now from overseas, the Army confirmed late Tuesday.

Michael J.L. Wilson, an assistant professor of political and cultural studies at the University of California Santa Barbara, said Mr. Lee was committed helping people in severe, life-defending conditions.

Mr. Wilson appeared on Fox News in 2014, drawing extensive social media coverage and the attention of politicians. He also wrote federal papers at a Congressional hearing, and he did so in a public May meeting with friends of the Falmouth Fish and Wildlife Society.

Mr. Lee's family said he was conscious and had taken part hospitalized treatment.

His attorney, Falmouth County, declined to comment.

Late Tuesday, officials on the health.gov website said the agency's programs in all 99 counties worked equally. The website highlighted the differing views on several crucial issues, as the events take on an expanded response to the country in the wake of the crisis.

"We have these people who are out in these rains and valleys in these deep jungles, these people in sinkholes, these people with cattle," one resident, C.R.M. Fort Lee, Ga, blamed the state the flooding. "And, we have internally displaced people, obviously. Hopefully those people have a better chance to get back on their feet at some point.

We know that some people have been featured in the movies or in the news while watching them. For example, a movie will show Captain William J. Jr. Hoyt (taken in a modern repellum meldass, photorealized) as he is told about his doctor: a former doctor, a depressive, and now one crucial witness he's lived with.

And if it doesn't just end with the character, Hoyt will spend months with Christopher Nolan as the titular, and — in yet another upcoming movie even more. George Reisingerly, director of the late John Segel and George Clo, has also directed Red-Hurts. The film is one of the most iconic Star Trek films of the 1960s, giving the Starfleet actor a better idea of who he is than most people realize. Over the weekend in Washington, he has found — not that he did, but than many would — that he will be replacing his star with the beginning of a larger group in him that includes World Health Organization's Director Tom Tomlinson.<|endoftext|>Word has bubbed in a broad swath of Washington political establishment, with one of the first time lawmakers criticized the Obama administration in the wake of an economic research study that that 400 million Americans die from heart attacks and birth defects.

"<|endoftext|>

Figure 12: A text sample generated by 8-step decoding with ReDi[3]

<|endoftext|>, in the NBA, football and Kurt Ramon on HBO's The Wire.

"It sends a message we're sending and we're sure that [the industry] doing their job." ()

Cameron is an actor and is one of the movie stars of the women's baseball union union (L.A). He will co-star Dustin Sheen and Ted Smith in The Jack Black, too. The movie will mark the 50th anniversary this year of the centennial of baseball. An Australian writer is writing the screenplay, the follow-up, the movie, and is making arrangements to make the film.

Cameron will see TOG: Batty time-traveling through sports, action, grills and music. The film was first made in 2003 - Australian release!

"In the movie, we really capture some of those spirit of L.A.. The film is populated, the players and the film stars stars as players, and it tells stories that will make people remember stories about sports.

"We use sports movies as a way to connect people, and we want them to remember what they do no matter how they finish. It is told through a female lens."

TAG: Batty is 0.4

"Women wake up now to read this movie, is is, very, very important news. Much research is important, and especially science fiction and requires great preparation preparation and effort.

"Men can make bad predictions, and so when it approach L.A. we should be careful and make sure that the industry are doing their job."<|endoftext|>When the U.S. was to be building the tallest building of the U.E. Building two centuries it, there was an online marketplace called Bazaar, which first was launched in 1994. The site since become a popular platform for investors in properties like Dr. Seuss, The Real Estate, and Justin Bieber, and it considered considered a boom time. Many other sites in the United States are offering the same deals to owners of government buildings for deals related to housing issues.

The site was a hassle to set up and promises to be a one-stop experience for all the journalists looking to get a good look at the major U.S. idea behind the site was built, specifically for those looking to make the switch between a movie with Simon Johnson, NBC News, a TV show, and a spaceship.

We'reShare our findings, can check out the website about the site here.

Of course, even Dr. Who is is famous when he asked:

How and where is he? Where where it all come from?

The Superman, R. Goldberg is the guy who directed the Batman movie Superman, and is also an actor. He is a well-known author of several films, Martin Gaerdes: The Return.

Goldberg, Mr. Supreme is a brilliant former Federal Reserve, and a former U.S.T. senator and Attorney-General of the U.S. He was a law and a field professor at Harvard in 1995. He graduated M.O. Harvard Harvard in 1998, and M.D. in 2000. He's degrees, one Johns Princeton University in 2001, and the Johns Hopkins University in 2002.

Other things to know about this site:

Flaset Tau: Mr. Oz, up together with a group of Bible Studies students in early June of this year,. J. Dozen, G(C-)formerly the family patriarch, is the fourth-Secretary of the United States. Fired by CBS in November 2014.

T-fus

Time: J.J. has made some of the most controversial decisions to right-wing politicians, including The Wolf of Meand, the president of the United States, in recent films.

Advertisement

Read the more on Slate: Michaela Pmana

Now, it lets you get all the latest information you need about the Star Trek universe. Later this week, you can expect to take part in the events of the 2017 Action Hero.

This is a great way so our fans can build on our work and invest in telling the real truth about the franchise. Let us turn all of the voting machines for your favorite TV series, and what could could possibly think to say? The original Star, A.V.

The following clip is a trailer created to coincide with the episode and the 33th anniversary of the 2017 Action Hero event.

The cast signed David Tennattis (91) and C.I.E. Shatner (R<|endoftext|>

Figure 13: A text sample generated by 8-step decoding with Di4C+ReDi[1]

