# OpenReview forum: "ReDi: Rectified Discrete Flow"
_NeurIPS.cc/2025/Conference — NeurIPS 2025 poster_

### Official Review · Reviewer_Ccyk · 2025-06-30

**Clarity:** 4
**Significance:** 3
**Originality:** 4
**Rating:** 5
**Confidence:** 3

**Summary:**

This paper proposes an iterative coupling rectification method for fast sampling in Discrete Flow Models (DFMs). Conventional DFM models factorize high-dimensional distributions by assuming independence across dimensions, which prevents them from capturing inter-dimensional dependencies. As a result, these methods suffer from accumulated approximation errors when using a small number of sampling steps. To address this issue, the authors propose to iteratively rectify the coupling, effectively reducing the Total Correlation (TC). Futhermore, the authors theoretically show that the CTC decreases monotonically with each rectification step. Experimental results demonstrate that the proposed method shows better performance than existing approaches in fewer sampling steps.

**Questions:**

* See weaknesses
* What is the computational cost of a single rectification step? A comparison of the overall training time with other baseline methods would help assess the efficiency of the proposed approach.

**Ethical Concerns:**

["NO or VERY MINOR ethics concerns only"]

**Final Justification:**

Since all the questions were raised during the rebuttal period, I will keep the initial rating.

**Limitations:**

yes

**Quality:**

3

**Strengths And Weaknesses:**

### Strength
* This paper is well-written and well-organized, with a clear motivation and a logical structure that is easy to follow.
* The theoretical analysis provides strong support for the proposed methodology. In the ablation study, it is empirically confirmed that CTC decreases on each rectification step.
* Compared to distillation-based approaches, which often involve complex architectures and loss functions, the proposed method is relatively simple.

### Weakness
* Details on the implementation of the rectification step (Eq. 6) and the training procedure with the new coupling are insufficiently described. Based on the experimental section, it appears that a distribution is computed by multiple sampling pairs, but this process is not clearly explained.

---

> ### Author Rebuttal · Authors · 2025-07-31
>
> We thank the reviewer for your valuable comments and positive assessment for our work. Below, we address the questions and concerns you have raised.
> > **Q1.** Details on the implementation of the rectification step (Eq. 6\) and the training procedure with the new coupling are insufficiently described. Based on the experimental section, it appears that a distribution is computed by multiple sampling pairs, but this process is not clearly explained.
> >
> **A1.** We appreciate the reviewer’s comment. To obtain the new coupling by Eq.6, we first sample the initial point from the prior distribution by $X_0\sim p(X_0)$ and apply the iterative denoising using the pre-trained model to obtain a sample from the terminal distribution by $X_1\sim p_\theta(X_1|X_0)$. The obtained pair $(X_0, X_1)$ defines the new coupling i.e., $(X_0, X_1)\sim \pi_{k+1}(X_0, X_1)$, which derives the new conditional transition probability in Eq.2. The model $p_\theta$ is then trained with the updated transition probability with the same denoising objective, which corresponds to a rectification step.
>
> We will add this detailed description to the preliminary section (Section 3.1) and link it to Eq. 6 to ensure clarity and completeness.
>
> > **Q2.** What is the computational cost of a single rectification step? A comparison of the overall training time with other baseline methods would help assess the efficiency of the proposed approach.
> >
> **A2.**  We appreciate the comment. While our rectification process requires sequential training, we clarify that each rectification step is much more efficient than distillation for the following reasons. First, as discussed in Section 4.4. and Figure 4(b) in the main paper, the rectification process can be trained with only a small portion of the entire training data (50K images vs. 1M full training data). As a result, it greatly reduces the cost of forwarding pre-trained models and convergence speed of each rectification step. Second, unlike distillation approaches, our method requires only the student model during training, avoiding the cost of operating two models simultaneously.
>
> **Table A** compares the training time of ReDi with the teacher model and other distillation approaches. It shows that the entire rectification process for ReDi$^3$-distill requires **only \~3% of the time** needed to train the initial teacher model, and faster training than distillation methods up to three rectification steps. We appreciate the reviewers’ comment and will incorporate this result in the paper.
>
> **Table A.**
> ||Iter.|GPU Hour / Iter.|Total Training Time|
> |:-|:-|:-|:-|
> |MaskGIT (Teacher model)|1|1800h|1800h|
> |SDTT|3|68h|204h|
> |Di4c|1|50h|50h|
> |ReDi$^2$|2|15h|30h|
> |ReDi$^3$-distill|3|15h|45h|

---

> > ### Comment · Reviewer_Ccyk · 2025-08-02
> >
> > Thank you for taking the time to respond to my review. All of my questions have been addressed, and I have no further questions.

---

> > > ### Author Response · Authors · 2025-08-04
> > >
> > > We would like to thank the reviewer once again for the constructive feedback and positive assessment. We truly appreciate it and will make sure to incorporate all the helpful discussions into our manuscript.

---

### Official Review · Reviewer_9bDF · 2025-07-02

**Clarity:** 3
**Significance:** 3
**Originality:** 3
**Rating:** 4
**Confidence:** 4

**Summary:**

Authors investigate the discrete flow-based model’s problem of approximation error due to factorization, which causes the inability of such models to generate data in a few-step manner. To overcome this problem, they propose and theoretically justify that iteratively applying rectification of the coupling minimizes conditional Total Correlation (TC) and consequently reduces the approximation error. Furthermore, they empirically demonstrate that approximated conditional TC strictly decreases over rectification iteration. Finally, authors validate the ability of rectified models in a few-shot generation. The results demonstrate noticeable qualitative improvements over distillation-based baselines in one-step generation and comparable improvements in few-step generation.

**Questions:**

- Why do you directly contrast the proposed method with distillation approaches? However, the approach aims to solve the problem more broadly by solving the factorization problem, which is more similar to methods mentioned in Minor Weaknesses.
- In checklist question 16, you mention using an LLM for a “proof sketch”. What do you mean by “sketch”? Specifically, is the proof in the Appendix considered a sketch proof?

**Ethical Concerns:**

["NO or VERY MINOR ethics concerns only"]

**Final Justification:**

The method is original, and the provided experimental support is sufficient. However, there are still some questions regarding the proof of the main theorem.

**Limitations:**

Yes

**Quality:**

3

**Strengths And Weaknesses:**

**Strengths**
- The way authors adapt the rectification procedure to achieve a few-step generation for discrete flow-based models is original and nontrivial.
- Concise theoretical preliminaries without unnecessary diffusion or flow model training details that seamlessly introduce the factorization problem;
- Multiple experiments cover the image and text domains;
- Theoretical (Theorem 1) and experimental (Section 4.4) evidence of decreasing conditional TC.

**Weaknesses**
- In Section 3.1 you adopt a unified viewpoint on flow-based models. It neatly lies with the perspective of the flow matching models. However, the diffusion models are typically presented using process-reversing formalism. To avoid confusion, it would be helpful (especially having only diffusion model experiments) to add references to a few works on diffusion bridges, such as, [1] or [2]. More importantly, such miss leaves outside an interesting connection of rectified flows to IMF procedure that solves Schrödinger Bridge problem [Appendix A.2, 3] with recent works on discrete space versions of IMF [4, 5].
- In the text experiments, the authors do not assess diversity of samples. This is crucial, since there is a known pitfall of generative perplexity: it can be artificially lowered through token repetition [6]. Particularly, reviewing samples in appendix it could be noticed that the higher ReDi iteration $k$ the more repetition appears in samples. For example, in Figure 13 line 1 “about about”, line 3 “p.m., p.m.”, line 8 “is is … what what”, etc. All these examples highlight the question: whether low generative perplexity values demonstrate overall quality improvements or are just the result of “hacking” the metric by the model.

[1]	Liu, Xingchao, Lemeng Wu, and Mao Ye. "Let us Build Bridges: Understanding and Extending Diffusion Generative Models." NeurIPS 2022 Workshop on Score-Based Methods.

[2]	Peluchetti, Stefano. "Non-denoising forward-time diffusions." arXiv preprint arXiv:2312.14589 (2023).

[3]	Kim, Beomsu, et al. "Simple reflow: Improved techniques for fast flow models."

[4]	Kim, Jun Hyeong, et al. "Discrete Diffusion Schrödinger Bridge Matching for Graph Transformation."

[5]	Ksenofontov, Grigoriy, and Alexander Korotin. "Categorical Schrödinger Bridge Matching."

[6]	Wang, Yequan, et al. "Perplexity from plm is unreliable for evaluating text quality." arXiv preprint arXiv:2210.05892 (2022).

**Minor Weaknesses**
- Do not mention some important references that cover the problem of the factorization [1, 2];
- In lines 273-274 it is quite hard to understand how Conditional TC is approximated; please provide some mathematical representation for better understanding;
- In the main theorem, I suppose there are missing assumptions, such as factorized conditional probability can’t be zero.

[1]	Liu, Anji, et al. "Discrete Copula Diffusion."

[2]	Xu, Minkai, et al. "Energy-based diffusion language models for text generation."

---

> ### Author Rebuttal · Authors · 2025-07-31
>
> We thank the reviewer for your valuable comments and positive assessment for our work. Below, we address the questions and concerns you have raised.
> > **Q1.** In Section 3.1 you adopt a unified viewpoint … such miss leaves outside an interesting connection of rectified flows to IMF procedure that solves SB problem \[Appendix A.2, 3\] with recent works on discrete space versions of IMF \[4, 5\].
> >
> **A1.** We thank the reviewer for these constructive comments and insightful suggestions. We agree that incorporating the suggested connections will significantly enrich the paper's context and clarify its position within the broader field.
>
> In the revised manuscript we will clarify the unified viewpoint in discrete generative modeling and discuss the connection to Schrödinger Bridge Problem (SBP) and Iterative Markovian Fitting (IMF). Specifically, we will highlight the procedural similarities between our ReDi method and the IMF algorithm, noting that our iterative training and data generation steps are analogous to the Markovian and reciprocal projections of IMF. We will also briefly introduce recent works that apply the SBP/IMF framework to discrete domains as the reviewer suggested.
> We believe these additions will provide valuable context and strengthen the paper's theoretical foundation. We thank the reviewer again for this valuable feedback, which has helped us improve our manuscript.
> > **Q2.** In the text experiments, the authors do not assess diversity of samples.This is crucial, since there is a known pitfall of generative perplexity.
> >
> **A2.** We thank the reviewer for the insightful comment. Upon the reviewer’s suggestion, we measured the unigram entropy alongside the Generative Perplexity (PPL), following the methodology in \[1\]. Aligned with the reviewer’s observation, we find that the entropy of the generated texts tends to decrease as we iterate more rectification steps. Upon closer inspection, we identified that this is inherited from the limited generative performance of the pre-trained model, perhaps attributed to its limited capacity (GPT-2-small backbone). Specifically, the generated distribution of the pre-trained model (PPL: 42.43 / entropy: 5.22) has a significant gap to the real data (PPL: 14.36, entropy: 5.44). When our rectification process is applied iteratively, this distributional gap can grow with each step, leading to low-diversity characteristics.
>
> We believe that employing higher-capacity pre-trained models can easily address this problem. To demonstrate this given a limited rebuttal timeframes, we conduct an experiment on a subset of OpenWebText with 1M tokens, ensuring that our teacher model has sufficient capacity to fit the distribution. As shown in **Table A, B**, in this setting, our method achieves significant PPL improvements in the few-step regime (2-8 steps) while simultaneously preserving high entropy. This demonstrates that when the base model is faithfully modeling data distribution, our method effectively improves few-step generation quality without sacrificing diversity.
>
> We deeply appreciate the reviewer’s comment and will update the experiments in the paper with larger pre-trained backbones and full OpenWebText dataset.
>
> **Table A.**
> |Gen.PPL|2|4|8|16|32|256|1024|
> |:-|:-|:-|:-|:-|:-|:-|:-|
> |Teacher|390.89|60.76|21.80|17.51|16.53|15.37|14.97|
> |ReDi$^1$|169.65|25.73|18.16|17.19|16.80|15.93|15.51|
> |ReDi$^2$|94.66|20.25|17.60|17.14|17.28|16.58|16.38|
> |ReDi$^3$|74.51|20.09|17.50|17.35|17.36|17.35|17.27|
>
> **Table B.**
> |Entropy|2|4|8|16|32|256|1024|
> |:-|:-|:-|:-|:-|:-|:-|:-|
> |Teacher|5.12|5.37|5.46|5.48|5.47|5.45|5.42|
> |ReDi$^1$|5.09|5.44|5.48|5.49|5.49|5.47|5.45|
> |ReDi$^2$|5.37|5.45|5.47|5.48|5.49|5.48|5.49|
> |ReDi$^3$|5.43|5.45|5.47|5.48|5.48|5.50|5.49|
>
> \[1\] Dieleman, Sander, et al. “Continuous diffusion for categorical data.”
>
> > **Q3**. Do not mention some important references that cover the problem of the factorization. \[1, 2\]; Why do you directly contrast the proposed method with distillation approaches?
> >
> **A3.** We thank the reviewer for suggesting these important references and for the insightful question regarding our choice of comparison.
>
> The main reason that we choose the distillation approaches as baselines is because they share the same goal and settings with our method: improving the few-step generation performance of the pre-trained discrete diffusion model. While the suggested papers (\[1, 2\]) are highly relevant for tackling the same factorization problem, they operate under different assumptions or with different goals, which makes a direct experimental comparison challenging:
> * Discrete Copula Diffusion \[1\], like our work, aims to solve the factorization problem for few-step generation. However, it assumes the availability of an additional pretrained autoregressive model for a copula model. Our experimental setup does not include such a component i.e., we aim to directly improve the few-step performance of pre-trained discrete diffusion models without external models
> * EDLM \[2\] reduces the approximation error using an energy-based approach. While this can reduce the number of decoding steps, it is not optimized for sampling efficiency. As detailed in their algorithm, each decoding step requires multiple evaluations of the energy-based model. This fundamentally differs from our goal of a truly efficient one- or few-step generative process.
> Therefore, we believe that distillation-based models, which share our specific goal of improving a base model's sampling efficiency without these additional components or computational costs, represent a more direct and appropriate comparison group for our work.
>
> Moreover, we agree that both papers are highly relevant as they address the same core problem of factorization error in discrete diffusion. We will add a detailed discussion of these works and a comparison of their methodologies to our Related Works section to better contextualize our contribution. We thank the reviewer again for these excellent suggestions.
>
> \[1\] Liu, Anji, et al. "Discrete Copula Diffusion."
> \[2\] Xu, Minkai, et al. "Energy-based diffusion language models for text generation."
>
> > **Q4.** In lines 273-274 it is quite hard to understand how Conditional TC is approximated; please provide some mathematical representation for better understanding.
> >
> **A4.** We thank the reviewer for this question and are happy to clarify our estimation procedure.
>
> We approximated the Conditional TC via Monte Carlo sampling, as the exact transition probability $p(X_1|X_0=x_0)$ is intractable. Our process involved generating 10 samples of $x_1$ for each of 5,000 initial states $x_0$. From these samples, we estimated the empirical conditional distributions ($\hat{p}(X_1|X_0)$ and $\hat{p}(X^i_1|X_0)$) by counting occurrences, which then allowed us to compute the Conditional TC. We will add this detailed description to the appendix.
>
> > **Q5.** In the main theorem, I suppose there are missing assumptions, such as factorized conditional probability can’t be zero.
> >
> **A5.** We thank the reviewer for this precise question regarding the assumptions for our main theorem.
> The reviewer correctly points out that the KL-divergence $D_{KL}(p||q)$ requires that $q$ must be non-zero on the support of $p$. In the context of the Conditional Total Correlation, $p$ corresponds to the conditional joint distribution $p(X_1|X_0)$ and $q$ corresponds to the factorized product of conditional marginals, $\prod_i p(X^i_1|X_0)$.
>
> However, we would like to clarify that no additional assumption is needed here. This is due to a fundamental property of probability distributions: for any given condition $X_0$, if a point $X_1$ has a non-zero probability under the conditional joint distribution (i.e., $p(X_1|X_0) > 0$), then each of its corresponding conditional marginal probabilities $p(X^i_1|X_0)$ must also be non-zero. Consequently, the factorized term $q = \prod_i p(X^i_1|X_0)$ is guaranteed to be non-zero on the support of $p(X_1|X_0)$. Therefore, the requirement for the KL-divergence is naturally satisfied without needing an explicit assumption in our theorem. We will add a footnote to the paper to make this point clear.
>
> > **Q6.** In checklist question 16, you mention using an LLM for a “proof sketch”. What do you mean by “sketch”? Specifically, is the proof in the Appendix considered a sketch proof?
> >
> **A6.** We apologize for the ambiguity. To be clear, the proof in the Appendix is a complete and formal proof, not a sketch. Our use of the term "proof sketch" in the checklist was intended to describe using the LLM for background research on mathematical tools (e.g., the Data Processing Inequality). The LLM did not write or generate the derivation, and the proof is our own work.

---

> > ### Comment · Reviewer_9bDF · 2025-08-05
> >
> > Thank you for your thorough comments. Upon revisiting the proof of Theorem 1, I began to question the equality stated after line 357. It appears not to be that obvious without a precise definition of the model’s conditional distribution $p_\theta(X_0, X_1)$. It seems that this equality may fail for some choices. Could you please provide a rigorous justification for why applying  $\Phi$ to $Q_k$ yields a factorized conditional distribution? I would greatly appreciate a clear explanation, as without it, my score may be negatively affected.

---

> ### Author Response · Authors · 2025-08-09
>
> We thank the reviewer for the question that leads us to clarify the theorem.
> We would like to recall Theorem 1 and provide a justification how applying $\Phi$ to $Q_k$ yields $Q_{k+1}$.
>
> **Theorem 1**. Let $\pi_k(X_0, X_1)$ be a coupling at iteration $k$, and let $\pi_{k+1}(X_0, X_1)=p(X_0)p_\theta(X_1|X_0)$ be the "rectified" coupling obtained via the ReDi procedure at iteration $k$. Then it satisfies the following:
>
> $$
> TC_{\pi_{k+1}}(X_1|X_0) \leq TC_{\pi_k}(X_1|X_0).
> $$
>
> In the proof we assumed that $p_\theta(X_1|X_0) = \prod_i p_\theta(X_1^i|X_0),$ rectifying the coupling with one-step decoding process. In this case, $\Phi(Q_k)=Q_{k+1}$ holds as:
>
> $$
> \Phi(Q_k)=E_{X_0, X_1}[p(X_0) \prod_{i=1}^N p_{\pi_k, 1|0}(X_1^i|X_0) \delta_{X_0}(X_0')p_\theta(X_1'|X_0)]\\
> $$
> $$
> =p(X_0')p_\theta(X_1'|X_0')\\
> $$
> $$
> =p(X_0') \prod_{i=1}^N p_\theta(X_1^i|X_0)\\
> $$
> $$
> =Q_{k+1}.
> $$
>
> While the proof is simple and general to other $f$-divergences that support data processing inequality, it is limited to one-step decoding process.
>
> ---
>
> Below, we provide an extended proof for more general M-step decoding processes, which is defined as $p_\theta(X_1|X_0)=\sum_{X_{t_1}, ... X_{t_{M-1}}}\prod_k\prod_ip(X_{t_{k+1}}^i|X_{t_k})$.
> The following properties are convenient:
>
> **Property 1. (Pythagorean Inequality for KL Divergence [1])** For a distribution $q$ in a log-convex set of distributions $Q$, if $q^* = \arg \min_{q\in Q} D_{KL}(p||q)$ and $r \in Q$, then
>
> $$
> D_{KL}(p||r) \ge D_{KL}(p||q^* ) + D_{KL}(q^*||r) \quad \text{(Eq.A)}
> $$
>
> ### Extended Proof of Theorem 1:
>
> We make two assumptions:
>
> * Assumption 1. Let $P$ be the family of M-step decoding processes. We assume that our model $p_\theta(X_1|X_0)$ lies within the log-convex hull of $P$.
>
> * Assumption 2. At each rectification step, our model is trained to minimize $D_{KL}(p_{\pi_k, 1|0}(X_1|X_0)||p_{\theta}(X_1|X_0))$. We assume that the model has reached the minimizer.
>
> Assumption 1 is plausible as $P$ becomes close to the hypothesis space of $p_\theta(X_1|X_0)$ for sufficiently large steps $M$.
> Assumption 2 is in line with step-wise optimality assumptions used in, e.g., GANs [2].
>
> The proof that $TC_{\pi_{k}}(X_1|X_0) \ge TC_{\pi_{k+1}}(X_1|X_0)$ proceeds as follows:
>
> $$
> TC_{\pi_{k}}(X_1|X_0) = E_{X_0}[D_{KL}(p_{\pi_k, 1|0}(X_1|X_0)||\prod_ip_{\pi_k, 1|0}(X_1^i|X_0))] \quad \text{(Definition of Conditional TC)}
> $$
> $$
> \ge E_{X_0}[D_{KL}(p_{\pi_k, 1|0}(X_1|X_0)||p_{\theta}(X_1|X_0))] + E_{X_0}[D_{KL}(p_{\theta}(X_1|X_0)||\prod_ip_{\pi_k, 1|0}(X_1^i|X_0))] \quad \text{(by Eq.A)}
> $$
> $$
> \ge E_{X_0}[D_{KL}(p_{\theta}(X_1|X_0)||\prod_ip_{\pi_k, 1|0}(X_1^i|X_0))] \quad (\text{since }D_{KL} \ge 0)\\
> $$
> $$
> =E_{X_0}[D_{KL}(p_{\pi_{k+1},1|0}(X_1|X_0)||\prod_ip_{\pi_k, 1|0}(X_1^i|X_0))] \quad \text{(by Eq.6 in the paper)}\\
> $$
> $$
> =E_{X_0}[D_{KL}(p_{\pi_{k+1},1|0}(X_1|X_0)||\prod_ip_{\pi_{k+1}, 1|0}(X_1^i|X_0))  + \sum_iD_{KL}(p_{\pi_{k+1}, 1|0}(X_1^i|X_0)||p_{\pi_{k}, 1|0}(X_1^i|X_0))] \\
> $$
> $$
> \ge E_{X_0}[D_{KL}(p_{\pi_{k+1},1|0}(X_1|X_0)||\prod_ip_{\pi_{k+1}, 1|0}(X_1^i|X_0))]  \quad (\text{since }D_{KL} \ge 0)
> $$
> $$
> = TC_{\pi_{k+1}}(X_1|X_0). \quad \square
> $$
>
> We sincerely thank the reviewer for their insightful feedback, which has helped us strengthen our proof. We will incorporate this extended version into the revised manuscript.
>
> [1] Wolfer, Geoffrey and Watanabe, Shun. "Geometric Aspects of Data-Processing of Markov Chains."
>
> [2] Goodfellow, Ian J., et al. "Generative adversarial nets."

---

### Official Review · Reviewer_EMvH · 2025-07-03

**Clarity:** 3
**Significance:** 4
**Originality:** 4
**Rating:** 5
**Confidence:** 3

**Summary:**

Discrete Flow-based Models (DFMs), while effective for high-quality discrete data generation, suffer from slow sampling speeds due to their reliance on iterative decoding processes. This paper first characterizes the approximation error from factorization using Conditional Total Correlation (TC), which quantifies interdimensional dependencies. Next, the authors propose Rectified Discrete Flow (ReDi), a novel iterative method that reduces factorization error by rectifying the coupling. ReDi theoretically guarantees a monotonic decrease in Conditional TC with each step and empirically demonstrates this reduction. Unlike traditional distillation methods that often involve complex multi-model training, ReDi simplifies the process by operating on a single DFM and directly rectifying the coupling, making it widely applicable and compatible with existing distillation frameworks. Empirical results show ReDi significantly improves few-step and one-step generation performance in image and text synthesis, outperforming or matching existing baselines.

**Questions:**

# Questions

1. Could the authors kindly define $\delta_{x_0}(x_t)$ in Eqn (1)? Do the authors mean something like $\delta_{x_0}(x_t) = \begin{cases}1 & \text{if } x_t = x_0 \\\\ 0 & \text{otherwise} \end{cases}$?
2. L168: How do you determine the stopping criteria for rectification? Do you measure some distribution-level metric like FID after each rectification? Or is K heuristically set to ~2 or ~3, following common practice in continuous rectified flows?

3. How do the time/compute requirements of ReDi compare with existing approaches?

4. Why are different CFG levels used for training ReDi$^1$+ReDi$^2$ vs ReDi$^3$-distill? (Appendix, L393)

**Ethical Concerns:**

["NO or VERY MINOR ethics concerns only"]

**Final Justification:**

Discrete diffusion/flow matching is increasingly becoming a topic of interest to the generative learning community. I believe this work further expands the field in a productive direction, by translating some ideas from classic rectified flows (i.e. reflow techniques) into the discrete domain.

I thus recommend accept.

**Limitations:**

Yes, addressed.

**Quality:**

4

**Strengths And Weaknesses:**

# Strengths

1. The paper is well-written. The proposed approach is well-motivated and important to the generative learning community.

2. The paper provides a way to characterize the factorization error in DFMs by using Conditional TC. This can be a useful tool in subsequent works in the field.

3. ReDi does not require maintaining a teacher and a student model unlike existing methods that follow a distillation paradigm, thereby easing memory requirements.

4. The authors demonstrate improvements in few-step and one-step generation performance across image and text synthesis tasks, outperforming or matching existing baselines, highlighting its practical effectiveness.

# Weaknesses

1. The paper could use some more exposition on the suitability of Conditional TC for characterizing factorization error (L130).

    - Are there any previous works that attempt to measure the gap between the conditional transition probability and the factorized product of marginals, or is this paper the first to do so?
    - Are there other metrics that can be used to characterize/measure factorization error? E.g. Cond. TC measures the expected KLD between the conditional and the marginal.
        - Could the authors briefly highlight why KL would be more appropriate than other methods in this scenario? (E.g. Cond. TC is directional, as $D_{KL}$(cond || factorized) $\neq$ $D_{KL}$(factorized || cond). Is this better than evaluating something symmetric like JS divergence?)
    - Some discussion like this would enrich the paper.

2. In Fig. 4a, the authors show that TC keeps decreasing with rectification iteration (shown up to 4). But it is noted in L277-279 that the authors empirically observe that the actual quality of the DFM drops with more ReDi iterations.
    - This empirical measurement seems to be absent; it could be shown alongside the TC in Fig. 4a.
    - Even though TC is decreasing, why does the performance of the model get poorer over time? Does that mean there is something (e.g. accumulated error) that TC fails to sufficiently capture? Some additional discussion on this phenomenon is required.

3. In L213, the authors mention they report the best FID score from trials with CFG values [1, 2, ..., 8]. Does each row in Table 1 correspond to a different CFG value? (and what are these values?) Does ReDi with different inference step count work better with different CFG values?
    - Since the authors have evaluated trials over a large set of CFG values, it would be useful if these results are provided as a curve.

4. Though ReDi's performance is outstanding for 1 step generation, for 4 step generation, it is mostly comparable or a little behind other methods. Do the authors have any intuition as to why?

---

> ### Author Rebuttal · Authors · 2025-07-31
>
> We thank the reviewer for your valuable comments and positive assessment for our work. Below, we address the questions and concerns you have raised.
>
> > **Q1.** Are there any previous works that attempt to measure the gap between the conditional transition probability and the factorized product of marginals?
> >
> **A1.** We thank the reviewer for the question. We interpret the term "measure the gap" as having two potential meanings—(1) to empirically quantify it, and (2) to theoretically characterize it with a mathematical concept. We will address both aspects below.
>
> Regarding the empirical measurement of this gap, to the best of our knowledge, our work is the first to propose a method to quantify it. We accomplish this by estimating the Conditional Total Correlation (CTC) via Monte Carlo sampling.
>
> Regarding the theoretical characterization, previous works have indeed conceptualized this gap using various metrics. For instance, Di4C \[1\] used Total Variation to represent it. We also acknowledge that Discrete Copula Diffusion \[2\] uses a concept related to CTC to characterize the factorization error of model. We would like to note that our key contribution is distinct in that we focus on the CTC of the data coupling itself (rather than the model), and introduce a rectification mechanism to empirically minimize this gap by updating the coupling.
>
> \[1\] Hayakawa, Satoshi, et al. “Distillation of Discrete Diffusion through Dimensional Correlations.”
> \[2\] Liu, Anji, et al. "Discrete Copula Diffusion."
>
> > **Q2.** Are there other metrics that can be used to characterize/measure factorization error? Why KL would be more appropriate than other methods?
> >
> **A2.** We thank the reviewer for this insightful question. We would like to answer it from a theoretical perspective and an empirical measurement perspective. We also wish to clarify that KL-divergence was not used as a direct training objective in our work, but rather as a tool for our theoretical and empirical analysis.
>
> From a theoretical perspective, the reviewer is correct that other metrics can be used. Any $f$-divergence is suitable for characterizing the factorization gap, as our Theorem 1 relies on the Data Processing Inequality (Appendix, L358-359), which holds generally for all $f$-divergences i.e., our rectification process reduces all $f$-divergences between the joint and factorized distributions, including Total Correlation.
>
> From an empirical measurement perspective, however, KL-divergence offers a crucial advantage in computational tractability. Our method for measuring the gap relies on Monte Carlo estimation. To compute the KL-divergence $D_{KL}(p||q)$, we only need to evaluate the term $p \log(\frac{p}{q})$ for the points sampled from the true distribution $p$. In contrast, estimating other f-divergences (such as JS-divergence) would require evaluating terms over the entire support of the factorized model $q$. In a high-dimensional discrete space, this is computationally infeasible.
>
> Therefore, while many metrics are valid in theory, KL-divergence was the practical and well-motivated choice for our empirical analysis. We will add this discussion to the appendix to clarify our reasoning.
>
> > **Q3.** Though ReDi's performance is outstanding for 1 step generation, for 4 step generation, it is mostly comparable or a little behind other methods. Is there something that TC fails to sufficiently capture?
> >
> **A3.** We thank the reviewer for this crucial question regarding the relationship between Conditional TC and generation quality. The full performance metrics are provided in **Table A**.
>
> **Table A**.
> ||TC of $\pi_{k}$|1-step FID|1-step IS|4-step FID|4-step IS|
> |:-|-|-|-|-|-|
> |MaskGIT|837|95.16|12|10.90|184|
> |ReDi$^1$|642|37.43|49|7.58|228|
> |ReDi$^2$|580|21.80|90|7.86|240|
> |ReDi$^3$|533|17.02|120|8.37|247|
>
> As shown in the table, we observe different trends in performance. In the multi-step regime, the 4-step FID begins to increase after $K=2$. In contrast, the 1-step FID consistently improves as TC decreases. This phenomenon arises because TC captures the factorization error but does not capture the discrepancy between our model's learned distribution, $p_\theta(X_1)$, and the true data distribution, $p(X_1)$. In an ideal scenario where the base model is perfect ($p_\theta = p$), each rectification step would improve generation quality. However, this discrepancy may exist in practice, meaning each rectification step can introduce a small distributional drift even as it reduces the factorization error.
>
> The net effect of this trade-off is highly dependent on the number of sampling steps. In the multi-step regime, the factorization error is already relatively small. Here, the marginal benefits of further TC reduction can be overshadowed by the accumulated distributional drift. In contrast, in fewer-step regimes (e.g., 1-step generation) where the initial factorization error is much larger, the significant performance gain from reducing TC far outweighs the minor impact of the drift. This explains the outstanding 1-step performance.
>
> We are grateful for this question and will add this detailed discussion and the corresponding data to the main paper.
> > **Q4.** Does each row in Table 1 correspond to a different CFG value?
> >
> **A4.** The reviewer is correct that we report the results with the best CFG value for each method in Table 1. To ensure reproducibility, we will add the below table in our appendix.
>
> **Table B.**
> |Step|Model|CFG|
> |:-|:-|:-|
> |1|MaskGIT|6|
> ||SDTT|5|
> ||Di4C|5|
> ||ReDi$^1$|3|
> ||ReDi$^2$|2|
> ||ReDi$^3$-distill|1|
> |4|MaskGIT|8|
> ||SDTT|4|
> ||Di4C|8|
> ||ReDi$^1$|6|
> ||ReDi$^2$|4|
> |8|MaskGIT|3|
>
> > **Q5.** Does ReDi with different inference step count work better with different CFG values?
> >
> **A5**. The reviewer is correct that the optimal CFG value varies with the number of inference steps. In response to the reviewer's request, we provide the full ablation study on this relationship below. This analysis will be added to the appendix as a curve, following the reviewer's helpful suggestion.
>
> **Table C.** ReDi$^1$
> |CFG|1|2|3|4|5|6|7|8|
> |:-|:-:|:-:|:-:|:-:|:-:|:-:|:-:|:-:|
> |1-step|55.35|39.33|**37.43**|39.62|42.46|45.27|47.75|49.90|
> |4-step|22.39|12.49|9.22|8.06|7.69|**7.58**|7.59|7.62|
>
> **Table D.** ReDi$^2$
> |CFG|1|2|3|4|5|6|7|8|
> |:-|:-:|:-:|:-:|:-:|:-:|:-:|:-:|:-:|
> |1-step|26.59|**21.80**|22.30|23.38|24.54|25.51|26.33|27.22|
> |4-step|11.47|8.68|8.03|**7.86**|7.86|7.90|7.92|7.95|
>
> > **Q6.** Could the authors kindly define delta function in Eqn(1)?
> >
> **A6.** We thank the reviewer for pointing out this omission. The \'$\delta_x(y)$\' in Equation (1) refers to the Kronecker delta function, defined as:
> $$
> \delta_x(y) = 1 \text{ if } x=y\text{, else } 0.
> $$
> We will add this definition to the manuscript.
>
> > **Q7.** How do you determine the stopping criteria for rectification?
> >
> **A7.** The number of rectification iterations ($K$) is determined empirically by evaluating the generation quality after each step. Specifically, we monitor both FID and IS for 4-step generation to find the best trade-off, as shown in **Table E**.
>
> **Table E.**
> |4-step|FID|IS|
> |:-|:-:|:-:|
> |MaskGIT|10.90|184|
> |ReDi$^1$|7.52|228|
> |ReDi$^2$|7.86|240|
> |ReDi$^3$|8.37|247|
>
> Based on the trade-off between FID and IS, we identified $K=2$ as a "sweet spot" that provides a large IS gain without a significant FID penalty on 4-step generation. Accordingly, we reported the scores for ReDi$^1$ and ReDi$^2$ as they represent the most compelling performance points.
>
> > **Q8.** How do the time/compute requirements of ReDi compare with existing approaches?
> >
> **A8.** We thank the reviewer for this important question regarding computational cost. While our rectification process requires sequential training, we clarify that each rectification step is much more efficient than distillation for the following reasons. First, as discussed in Section 4.4. and Figure 4(b) in the main paper, the rectification process can be trained with only a small portion of the entire training data (50K images vs. 1M full training data). As a result, it greatly reduces the cost of forwarding pre-trained models and convergence speed of each rectification step. Second, unlike distillation approaches, our method requires only the student model during training, avoiding the cost of operating two models simultaneously.
>
> **Table F** compares the training time of ReDi with the teacher model and other distillation approaches. It shows that the entire rectification process for ReDi$^3$-distill requires **only \~3% of the time** needed to train the initial teacher model, and faster training than distillation methods up to three rectification steps. We appreciate the reviewers’ comment and will incorporate this result in the paper.
>
> **Table F.**
> ||Iter.|GPU Hour / Iter.|Total Training Time|
> |:-|:-|:-|:-|
> |MaskGIT (Teacher model)|1|1800h|1800h|
> |SDTT|3|68h|204h|
> |Di4c|1|50h|50h|
> |ReDi$^2$|2|15h|30h|
> |ReDi$^3$-distill|3|15h|45h|
>
> > **Q9.** Why are different CFG levels used for training ReDi$^1$+ReDi$^2$ v.s. ReDi$^3$-distill?
> >
> **A9.** The CFG levels to define rectified coupling was determined empirically, and our ablation studies show it differs for multi-step rectification versus one-step distillation. Specifically, lower guidance (CFG 1-2) was optimal for direct multi-step sampling (**Table G**), whereas strong guidance (CFG 8) was required to generate the best coupling for the one-step distillation task (**Table H**). This analysis will be added to the appendix for clarity.
>
> **Table G.**
> |ReDi$^1$ (4-step)|FID|IS|
> |:-:|:-:|:-:|
> |CFG 0|9.76|163|
> |CFG 1|7.52|228|
> |CFG 2|7.77|252|
> |CFG 3|8.63|283|
>
> **Table H.**
> |ReDi$^3$-distill (1-step)|FID|IS|
> |:-:|:-:|:-:|
> |CFG 1|14.11|139|
> |CFG 2|13.25|150|
> |CFG 8|11.68|182|

---

> > ### Comment · Reviewer_EMvH · 2025-08-02
> >
> > I thank the authors for their strong response, particularly for clarification regarding the empirical choice of CFG values. My concerns have been addressed; at this point I have no additional questions.

---

> > > ### Author Response · Authors · 2025-08-04
> > >
> > > We would like to thank the reviewer once again for the constructive feedback and positive assessment of our work. We truly appreciate it and will make sure to incorporate all the helpful discussions into our manuscript.

---

### Official Review · Reviewer_riBK · 2025-07-05

**Clarity:** 3
**Significance:** 3
**Originality:** 3
**Rating:** 4
**Confidence:** 3

**Summary:**

This paper addresses a key inefficiency in discrete flow-based generative models: the reliance on many sampling steps due to weak coupling between base and target distributions. While rectified flows have been used in continuous domains to straighten the path between source and target distributions, they have not been explored for discrete data. Moreover, most discrete flow models assume factorized transitions of the form $p(x_s \mid x_t) = \prod_i p(x_s^i \mid x_t)$, which ignore inter-variable dependencies and thus lead to suboptimal approximations. To overcome this, the authors propose ReDi---a progressive coupling mechanism. At each iteration $k$, the model samples from $p(x_0)$ and trains a model to predict $p_\theta(x_{k+1} \mid x_k)$. The authors demonstrate that this strategy contracts the factorization error, defined as the KL divergence between the joint and factorized distributions. Empirical results on ImageNet (class-conditional) and OpenWebText show that ReDi achieves one-step generation with competitive results compared to distillation-based methods.

**Questions:**

Questions and Comments: Can this progressive coupling mechanism be extended to continuous flows? Why was the discrete setting prioritized? How does generation quality (e.g., FID for images) improve compared to the baseline factorized model in the multi-step regime? What is the actual training cost versus knowledge distillation? Finally, how sensitive is performance to the number of coupling iterations $K$? Could $K$ be adaptively chosen?

**Ethical Concerns:**

["NO or VERY MINOR ethics concerns only"]

**Quality:**

2

**Strengths And Weaknesses:**

Strengths: The paper tackles a timely and nontrivial challenge in discrete generative modeling. The proposed progressive coupling mechanism is intuitive and theoretically well-grounded. The contraction result on the factorization error is an insightful contribution. The empirical results, especially one-step sampling, are promising.

Weaknesses: Training ReDi appears to require multiple forward models, trained sequentially (at least 4 iterations as shown in Fig. 4a), which raises concerns about total training cost. It’s unclear whether this method is actually more computationally efficient than distillation. Moreover, improvements in generation quality appear modest, and the method has not yet demonstrated scalability to larger datasets.

---

> ### Author Rebuttal · Authors · 2025-07-31
>
> We thank the reviewer for your valuable comments and positive assessment for our work. Below, we address the questions and concerns you have raised.
>
> > **Q1.** What is the actual training cost versus knowledge distillation?
> >
> **A1.** We appreciate the comment. While our rectification process requires sequential training, we clarify that each rectification step is much more efficient than distillation for the following reasons. First, as discussed in Section 4.4. and Figure 4(b) in the main paper, the rectification process can be trained with only a small portion of the entire training data (50K rectification data vs. 1M full training data). As a result, it greatly reduces the cost of forwarding pre-trained models and convergence speed of each rectification step. Second, unlike distillation approaches, our method requires only the student model during training, avoiding the cost of operating two models simultaneously.
>
> **Table A** compares the training time of ReDi with the teacher model and other distillation approaches. It shows that the entire rectification process for ReDi$^3$-distill requires **only \~3% of the time** needed to train the initial teacher model, and faster training than distillation methods up to three rectification steps. We appreciate the reviewers’ comment and will incorporate this result in the paper.
>
> **Table A.**
> ||Iter.|GPU Hour / Iter.|Total Training Time|
> |:-|:-|:-|:-|
> |MaskGIT (Teacher model)|1|1800h|1800h|
> |SDTT|3|68h|204h|
> |Di4c|1|50h|50h|
> |ReDi$^2$|2|15h|30h|
> |ReDi$^3$-distill|3|15h|45h|
>
> > **Q2.** Moreover, improvements in generation quality appear modest.
> >
> **A2.** We would like to kindly remind the reviewer that our primary goal is to improve the few-step generation performance by reducing the factorization errors in discrete diffusion models. As shown in Table 1 and discussed in Section 4.2, our method achieves significant improvements in the one-step setting compared to both the teacher model and other distillation baselines. While it shows comparable 4-step performance to competing methods, our method still exhibits much lighter memory for training compared to other distillation baselines that require additional teacher networks, and faster convergence using only a small portion of data for rectification (Fig 4(b) in the paper. please also see A1).
>
> > **Q3.** The method has not yet demonstrated scalability to larger datasets.
> >
> **A3.** We would like to clarify that the ImageNet and OpenWebText are widely used datasets to evaluate the image and discrete diffusion models. Although time and computational constraints prevented us from testing on larger datasets during the rebuttal phase, we believe that our method can easily scale to larger models or datasets for following reasons. First, our rectification process and its theoretical foundation are model-agnostic and are applicable on top of pre-trained discrete diffusion/flow models agnostic to their architecture and scale. Second, unlike distillation approaches that require both teacher and student models during training, our method does not increase the memory footprint beyond that of the original pre-training. We plan to open source our code to facilitate the future community efforts in scaling our approach.
>
> > **Q4.** Can this progressive coupling mechanism be extended to continuous flows? Why was the discrete setting prioritized?
> >
> **A4.** We would like to clarify the connection of our work to continuous flows and the motivation for prioritizing the discrete setting.
>
> Although similar progressive refinement mechanisms have been explored in the continuous domain, as we note in our Related Works (Sec. 2.3), our work was prioritized for the discrete setting because the core challenges are fundamentally different. For continuous flows, the primary obstacle to few-step generation is the non-straightness of probability paths. In discrete flows, however, the concept of path straightness is not well-defined, as there are no intermediate states between distinct categories. The critical bottleneck we address is instead the factorization error inherent to these models.
>
> Our key contribution is thus to characterize this specific factorization error and resolve it with our proposed rectification process. Interestingly, this process, despite being derived specifically to address the unique problem of discrete factorization, closely resembles the rectification methods used for the different challenges in continuous flows. We will add a discussion of this point to our method section to further clarify this distinction and connection.
>
> > **Q5.** How does generation quality (e.g., FID for images) improve compared to the baseline factorized model in the multi-step regime? Finally, how sensitive is performance to the number of coupling iterations $K$? Could $K$ be adaptively chosen?
> >
> **A5.** We would like to kindly clarify our model's performance in the multi-step regime and its sensitivity to the number of coupling iterations, $K$.
>
> Regarding the generation quality versus the baseline in the multi-step regime, our method exhibits a trade-off between FID and IS, as shown in **Table B**. While the Inception Score improves compared to the baseline, the FID moderately degrades as $K$ increases. It is important to note that our primary contribution lies in the few-step regime, where the model shows significant gains from the rectification as demonstrated in Table 1\.
>
> **Table B.**
> |8-step|FID|IS|
> |:-|:-:|:-:|
> |MaskGIT|6.51|227|
> |ReDi$^1$|6.57|249|
> |ReDi$^2$|7.37|261|
>
> Concerning the sensitivity to $K$, this behavior can be understood as a trade-off between two competing factors. On one hand, each rectification process reduces the Conditional Total Correlation (TC), which significantly improves performance with fewer-step regimes where the initial factorization error is large. On the other hand, each rectification can introduce a small amount of distributional drift due to imperfections in the base model. In the multi-step regime, this accumulated drift can offset the benefits of TC reduction, leading to the observed performance changes.
>
> Finally, regarding the adaptive selection of $K$, we agree with the reviewer that this is a viable approach. The general principle would be to monitor a chosen performance metric, such as FID or IS, after each rectification iteration and select the $K$ that provides the most desirable outcome based on that evaluation.

---

> > ### Comment · Reviewer_riBK · 2025-08-06
> >
> > Thanks for addressing my comments. I’ll keep my score as is. I recommend that the authors include a discussion of the limitations, such as the trade-offs in perceptual quality (e.g., FID), in the final version. It would also be helpful to analyze the sensitivity to the choice of K and clarify in what sense the method is lighter than distillation.

---

### Note · Authors · 2025-08-13

We sincerely thank all Reviewers, ACs, SACs, and PCs for their valuable and insightful feedback, which has been instrumental in strengthening our work.

We are encouraged that the reviewers found our method to be theoretically well-grounded (riBK, EMvH, 9bDF, Ccyk), supported by an empirical analysis on Conditional Total Correlation (TC) (riBK, EMvH, Ccyk). We also appreciate that the reviewers recognized the method as simple (Ccyk) and original (9bDF), and found its practical benefits in improving few-step and one-step generation (riBK, EMvH) and memory efficiency (EMvH).

---

We appreciate the reviewers' insightful questions. Our rebuttal discusses all concerns raised, with particular attention to the following points:
- **Performance over multiple iterations** (riBK, EMvH): Provided a quantitative analysis of the FID-IS trade-off across ReDi iterations.
- **Training cost** (riBK, EMvH, Ccyk): Included a training time comparison with baselines to demonstrate our method's efficiency.
- **Text diversity assessment** (9bDF): Included a quantitative assessment of text diversity with further analysis.
- **Relation to other frameworks** [1,2] (riBK, 9bDF): Added a discussion on the connections to rectified flows and iterative Markov fitting.
- **Comparison to recent work on factorization error** [3,4] (9bDF): Expanded the discussion to better contextualize our method.
- **Clarity of details** (EMvH, 9bDF, Ccyk): Provided clarifications on experimental details and the proof of Theorem 1.

---

Our detailed responses have addressed the concerns of Reviewer riBK, EMvH, and Ccyk. For Reviewer 9bDF's concern about the justification of Theorem 1, we have provided an additional justification and an extended proof. Although we have not yet confirmed if this fully resolves the reviewer's concern, we are hopeful that our explanation is sufficient.

We are grateful for the reviewers' constructive guidance. We believe the resulting discussions and planned revisions will significantly strengthen the manuscript, and we will incorporate these new results and clarifications into the paper.

[1] Liu, Xingchao, et al. "Flow straight and fast: Learning to generate and transfer data with rectified flow."

[2] Kim, Beomsu, et al. "Simple reflow: Improved techniques for fast flow models."

[3] Liu, Anji, et al. "Discrete Copula Diffusion."

[4] Xu, Minkai, et al. "Energy-based diffusion language models for text generation."

---

### Decision · Program_Chairs · 2025-09-17

**Decision:**

Accept (poster)

**Comment:**

This paper introduces ReDi, a rectified discrete flow framework that reduces factorization error via progressive coupling, supported by a theoretical contraction result and empirical validation. Reviewers appreciated the originality of extending rectified flows to the discrete domain, the clarity of exposition, and strong one-step generation performance. Main concerns include unclear efficiency trade-offs compared to distillation, modest gains in multi-step settings, limited evaluation of text diversity, and theoretical assumptions in Theorem 1. The rebuttal provided additional analyses (training cost, FID–IS trade-offs, text diversity metrics, extended proof) that addressed most concerns. Reviewer 9bDF maintained that the proof remains incomplete under current assumptions, but acknowledged that the method is empirically strong and recommended acceptance with the caveat that the theorem be reframed as a proposition if a strict proof cannot be provided. Overall, the work represents a novel and technically solid contribution that advances discrete generative modeling and demonstrates practical promise.